# How to Train Long-Context Language Models (Effectively)

## Abstract

We study continued training and supervised-fine-tuning (SFT) of a language model (LM) to make effective use of long-context information. We first establish a reliable evaluation protocol to guide model development—instead of perplexity, we use a broad set of long-context tasks, and we evaluate models after supervised fine-tuning (SFT) with instruction data as this better reveals long-context abilities. Supported by our robust evaluations, we run thorough experiments to decide the data mix for continued pre-training, the instruction tuning dataset, and other design choices such as position extrapolation. We find that (1) code repositories and books are excellent sources of long data, but it is crucial to combine them with high-quality short data; (2) training with a sequence length beyond the evaluation length boosts long-context performance; (3) for SFT, using only short instruction datasets yields strong performance on long-context tasks. Our final model, **ProLong-8B**, which is initialized from Llama-3 and trained on 40B tokens, demonstrates state-of-the-art long-context performance among similarly sized models at a length of 128K, outperforming Llama-3.1-8B on the majority of long-context tasks despite having seen 5% as many tokens during long-context training. Additionally, ProLong can effectively process up to 512K tokens, one of the longest context windows of publicly available LMs.[1]

> **Takeaways for continued training of long-context models**
>
> - **Evaluation** (§2): We target a range of long-context downstream tasks instead of perplexity or needle-in-a-haystack, while checking if the short-context performance is preserved. We evaluate models after SFT, which produces a clearer signal on long-context tasks.
> - **Data engineering** (§3): We conduct a series of ablations at a 5B-token scale. We find that using code repositories and long books as long-context data and mixing them with high-quality short-context data is crucial for both long-context performance and retaining the short-context capabilities of the pre-trained model.
> - **Scaling the data and the length** (§4): We scale up the training to 20B tokens at a 64K training length and 20B tokens at a 512K training length. Surprisingly, training on contexts longer than the evaluation length yields additional benefits.
> - **Supervised fine-tuning** (§5): We find that SFT with standard, short-context instruction datasets is sufficient for achieving good performance. Contrary to previous study, long synthetic instruction data does not boost the result in our setting.
> - **ProLong models** (§6): We present our final recipe and evaluation results here. All our code, data, and models will be made publically available.

## 1 Introduction

The ability of language models (LMs) to process extremely long inputs (for example, 128K tokens) has enabled new applications, such as book summarization or learning new tasks on the fly from many

---

[1]All our code, data, and models will be made publically available.

examples. However, adapting LMs to process long contexts is challenging from an infrastructure and data perspective, and many design decisions are not well understood by open-source practitioners.

While many works have focused on extending the context length of pre-trained LMs with minimal training (Chen et al., 2023; Peng et al., 2024), Fu et al. (2024) find that the above methods cannot even perform the simple needle-in-a-haystack (NIAH; Kamradt, 2024) task and it is necessary to continually train the LM on long documents for billions of tokens. Frontier open-source models, such as Llama-3.1 (Dubey et al., 2024) and Jamba (Lenz et al., 2024), also adopts a long-context continued training stage, followed by supervised fine-tuning (SFT) on instruction data. We adopt the same setting and study continued training and SFT of a pre-trained LM for effective long-context use.

We first establish a reliable evaluation protocol to provide meaningful signal for model development. Most existing work relies on either perplexity or NIAH for ablating training recipes. We demonstrate that neither is robust for guiding the development and opt for a broad range of downstream applications, such as retrieval-augmented generation (RAG), long-document summarization, and many-shot in-context learning (ICL). We also conduct our evaluations after performing SFT, as we observe that, on some long-context tasks, performance gains only emerge after SFT.

Guided by our evaluation protocol, we run comprehensive experiments with Llama-3-8B (8K original context window; Dubey et al., 2024) to study each component of long-context continued training, including data mixture, data and length scaling, supervised fine-tuning, and many other design choices such as position extrapolation. Many of our findings are surprising or contradictory to existing claims, for example, (1) training only on long data hurts long-context performance, (2) training on longer sequences than the evaluation length helps, and (3) SFT on only short instruction data is sufficient for good long-context performance. We outline our main takeaways and the structure of the paper in the takeaway box at the beginning of this section.

Our final model, **ProLong**, achieves the best performance at a 128K context length among 10B-parameter models, while taking only $5\%$ of the data budget compared to Llama-3.1's long-context training (Dubey et al., 2024). ProLong has a maximum context length of 512K tokens, making it one of the longest-context LMs available.[2]

## 2  GUIDE MODEL DEVELOPMENT WITH MEANINGFUL EVALUATIONS

A pre-requisite for training a strong LM is having a robust evaluation suite that can guide model development while tracking its utility in real-world applications. While synthetic benchmarks like needle-in-a-haystack (NIAH; Kamradt, 2024) and RULER (Hsieh et al., 2024) have gained much popularity due to their simplicity and controllability, we are interested in a wider range of tasks that reflect practical usage, such as the ability to reason over the whole document. In the following, we describe our evaluation protocols and showcase why they are critical to our model development.

### 2.1  EVALUATE ON DIVERSE AND REALISTIC TASKS

We first make the decision to use HELMET (Yen et al., 2024b) as our main evaluation suite, as it is one of the most comprehensive long-context benchmarks, covering the following task categories:

- **Recall**: Given a JSON file with random key-values pairs, retrieve the value for a specific key.
- **RAG**: Answer a question given many retrieved Wikipedia documents (*NQ*, *HotPotQA*, *PopQA*).
- **Re-rank**: Produce a top 10 ranking from a long and shuffled list of documents (*MSMARCO*).
- **ICL**: Learn classification tasks from many in-context examples, where the number of classes ranges from 6 to 151; average of 5 datasets (*TREC coarse/fine*, *NLU*, *Banking77*, *Clinc-150*).
- **QA**: Answer a question given a full-length book (*NarrativeQA*).
- **Summarization**: Summarize long legal documents (*Multi-LexSum*).

Overall, these diverse tasks reflect a range of long-context abilities including recall, reasoning, learning from context, and robustness to noisy inputs. Yen et al. (2024b) also show that HELMET produces model performance trends that are more consistent with human perceptions unlike other long-context benchmarks.

---

[2]Throughout the paper, we use binary prefixes K$= 2^{10}$, M$=2^{20}$, and B$=2^{30}$.

We showcase the importance of a robust evaluation suite in Table 1. As a predecessor of our work, Fu et al. (2024) only consider needle-in-a-haystack (NIAH) and perplexity during model development; evaluations on 3 tasks from HELMET reveal major short-comings of their model. We also see how NIAH and even the HELMET recall task become saturated for strong models (Llama-3.1-8B vs. 70B) while other task categories continue to detect differences in their long-context abilities.

Table 1: HELMET tasks offer a more holistic view of long-context abilities.

| Models | NIAH | HELMET | | |
| | | Recall | RAG | Re-rank |
|---|---|---|---|---|
| Fu et al. (2024) | 86.0 | 33.0 | 46.6 | 7.7 |
| Llama-3.1-8B | 100 | 98.7 | 62.8 | 26.6 |
| Llama-3.1-70B | 99.7 | 98.5 | 65.9 | 39.4 |

We offer more details about the HELMET evaluation, including its careful choice of metrics, in §B.1. If not otherwise specified, we average the performance for each category over all datasets and over evaluation lengths of 32K and 64K; for the final long-context score, we macro-average all categories.

**Why not perplexity?** Besides synthetic recall tasks, many previous works rely on perplexity (PPL) for evaluating long-context extensions of LMs (Chen et al., 2023; Fu et al., 2024; Lu et al., 2024), which is commonly measured on the PG19 books dataset (Rae et al., 2020). We use the ablation experiment from §3.2 to showcase why perplexity is not an indicative metric for developing long-context models. The experiment studies how the ratio of long documents affects the performance. We report both our evaluation and the perplexity measured on the last 32K tokens of 64K-length documents from PG19. As shown in Figure 1, while using more long data continues to improve PPL, it is clear that using 100% long data significantly hurts downstream long-context performance.

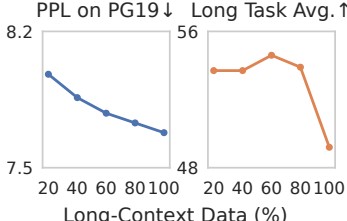

Figure 1: Making design decisions based on perplexity (PPL) is not optimal for long-context downstream tasks.

## 2.2 EVALUATE AFTER SUPERVISED FINE-TUNING

Supervised fine-tuning (SFT; Ouyang et al., 2022) is an additional training stage that fine-tunes the model on a small amount of natural-language instructions and corresponding responses; it enables a base LM to address user queries in a chat format and has become a standard step for producing frontier LMs. Here, we consider the difference between evaluating a model *before* or *after* SFT.

In preliminary experiments, we continue training Llama-3-8B-Base on 5B-token subsets from the data mix by Fu et al. (2024). The mix is based on SlimPajama (Soboleva et al., 2023) and upsamples long documents to constitute roughly 70% of tokens, while retaining the original domain proportions. Then we conduct SFT on several intermediate checkpoints with UltraChat (Ding et al., 2023).

We show the benchmarking results before and after SFT in Figure 2. Long-context evaluation shows clearer signals when it is conducted after SFT: (1) SFT shows that the model continues to improve with more training tokens on RAG and re-ranking, while the improvement is less clear or does not exist when evaluated before SFT. (2) SFT enables evaluation on realistic applications like QA and summarization, which require instruction following and have low performance before SFT. We also note that the variance from two random training runs is not substantially higher after the additional SFT phase. Therefore, unless otherwise specified, we report the long-context performance *after* SFT.

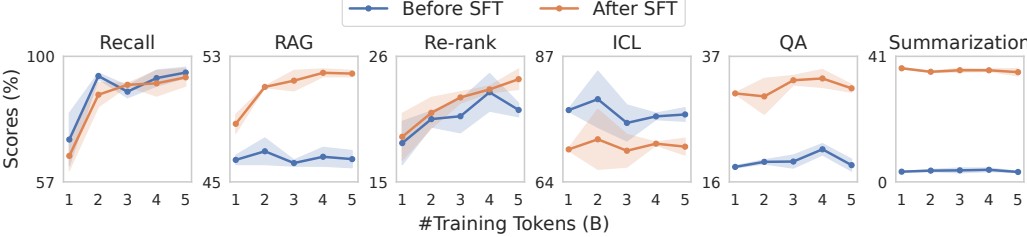

Figure 2: Improvements on RAG and re-ranking tasks are only observed when evaluating models after a supervised fine-tuning (SFT) phase on instruction data. The models are trained on the pre-training data mix by Fu et al. (2024). We report the mean and standard deviations over two training runs.

We dive deeper into supervised fine-tuning in §5 and explore different training datasets, as well as the use of synthetic long instruction data. However, we find that simply fine-tuning on UltraChat remains a surprisingly competitive choice.

## 2.3 CHECK THAT SHORT-CONTEXT PERFORMANCE IS PRESERVED

Long-context abilities should not come at the expense of short-context performance, particularly since short-context evaluations cover a wider range of capabilities, e.g., world knowledge, commonsense, and mathematical reasoning. However, short-context evaluation has largely been neglected by previous long-context research. We report on 5 tasks from the the Open LLM Leaderboard (Beeching et al., 2023): HellaSwag (Zellers et al., 2019), MMLU (Hendrycks et al., 2021), ARC-challenge (Clark et al., 2018), WinoGrande (Sakaguchi et al., 2021), and GSM8K (Cobbe et al., 2021). We evaluate short-context performance *before* SFT, since this allows for a direct comparison to the base model which was used as initialization for the long-context training.

**Previous techniques deteriorate short-context performance.** We show in Table 2 that both training-free position extrapolation, as well as fine-tuning with an existing long data mixture (Fu et al., 2024) do not preserve the strong performance of Llama-3-8B on standard short-context tasks. This motivates us to find data sources which retain the initial model's strong short-context performance.

Table 2: Applying position extrapolation (PE) to Llama-3-8B by changing the RoPE frequency base (§C.1) or fine-tuning it on a long-context SlimPajama mixture (Fu et al., 2024; Soboleva et al., 2023) deteriorates the performance of this top-shelf pre-trained LM on short-context tasks.

|  | HSwag | MMLU | ARC-c | WG | GSM8K |
|---|---|---|---|---|---|
| *Llama-3-8B* | 82.1 | 66.5 | 59.4 | 77.1 | 44.7 |
| + PE | 81.5 | 64.7 | 58.1 | 75.5 | 40.1 |
| + SlimPajama | 81.0 | 63.1 | 57.8 | 75.1 | 40.6 |

## 3 LONG-CONTEXT DATA CURATION

The quality and composition of training data has been found to be the most important factor for LM pre-training (Penedo et al., 2023; Wettig et al., 2024; Li et al., 2024a) and is therefore a primary focus of our study. To make data decisions, we perform ablation experiments: we continue to train Llama-3-8B-Base for 5B tokens with a maximum length of 64K tokens and evaluate according to §2. See §B.4 for more details of our ablation setting.

We aim to boost the long-context task performance while preserving the short-context performance of the original model. Starting from the intuition that the data should be a mixture of long and short documents, we study these choices separately. In our ablations, the long data is comprised of single-document chunks of 64K tokens, whereas for the short data, we construct batches by packing documents until we reach 64K tokens per sequence.

## 3.1 CODE REPOSITORIES AND BOOKS ARE GOOD SOURCES OF LONG-CONTEXT DATA

**SlimPajama.** We analyze the quantity of long data in SlimPajama (SP; Soboleva et al., 2023). Table 3 shows that books account for the majority of long-context tokens. When inspecting the long data in CommonCrawl (CC), we observe that though varied in quality, it also contains some book-like content, which future work could identify via data selection methods.

**Code repositories.** While only few files from GitHub reach a very long length (which also tend to be lower quality as suggested by Singh et al., 2024), we construct an abundant source of long-context data from the Stack (Kocetkov et al., 2023) by concatenating all files from a repository to form a single document. Unlike Guo et al. (2024), we do not order the files based on dependencies, which should increase the distance between dependent files and reduce recency bias.

Table 3: Long text documents ($\geq$64K tokens) by data sources.

| Data | #Long tokens |
|---|---|
| **Code Repos** | **98.8B** |
| **SP/Books** | **33.2B** |
| SP/CC | 15.3B |
| SP/Arxiv | 5.2B |
| SP/GitHub | 2.8B |
| SP/Wiki | 0.1B |
| SP/StackEx | <0.1B |
| SP/C4 | <0.1B |

Table 4: Impact of different long data sources, while keeping the 40% short data component fixed. Code repositories particularly helps long-context recall, while books are more effective on re-ranking, ICL, and summarization. Mixing the two sources achieves the overall best performance.

| Long Data (60%) | Long-Context | | | | | | | Short-Context |
|---|---|---|---|---|---|---|---|---|
| | Recall | RAG | Re-rank | ICL | QA | Summ. | Avg. | Avg. |
| CommonCrawl | 84.1 | 53.3 | 28.1 | 67.5 | 35.2 | 37.0 | 50.9 | 66.5 |
| Books | 94.9 | 53.9 | **30.7** | 72.2 | 33.2 | 37.7 | 53.8 | 65.5 |
| Code Repos | **99.2** | 53.8 | 29.0 | 61.2 | 34.7 | 36.2 | 52.3 | 65.9 |
| Books/Repos 1:1 | 96.0 | **54.9** | 29.4 | **73.9** | **35.7** | **37.9** | **54.6** | 65.5 |

**Data mixture.** We train models with 60% of long-context data and 40% of our ShortMix (§3.3). Table 4 shows that using code repositories alone performs best on stress-test recall tasks. Meanwhile, books are more broadly beneficial for in-context learning, summarization and re-ranking. An equal mix of books and code repositories achieves the best overall performance. Note that short-context task performance remains consistent due to our high-quality short data mix.

## 3.2 TRAINING ONLY ON LONG DATA HURTS LONG-CONTEXT PERFORMANCE

The ratio between short/long data is another crucial factor for downstream performance. Prior work either trains only on long data (Peng et al., 2024) or adds some short training data (Yen et al., 2024a; Fu et al., 2024). However, we are the first to systematically study the impact of short/long ratio.

Figure 3 shows that short task performance monotonically decreases as the long data increases. The trends for long-context vary by tasks and are further complicated by SFT: On tasks like recall and RAG, the performance before SFT prefers high proportions of long data, while the performance after SFT drastically deteriorates with more long data. We hypothesize that specializing the model only on long data makes it a poor initialization for generic SFT—highlighting the importance of evaluating checkpoints after SFT (§2.2). While some long-context tasks benefit from more long data consistently (ICL) or show no clear pattern (re-ranking), the best average performance is achieved at 60% long data and 40% short data, which we adopt for our final ProLong model.

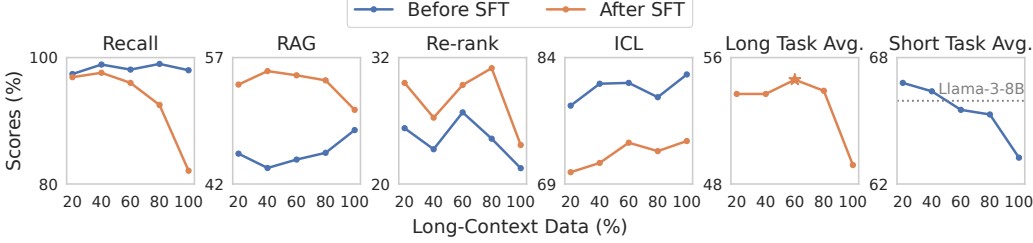

Figure 3: Impact of short/long data ratio. All models are trained on books/repos long data and our ShortMix for 5B tokens. More long data initially improves long-context performance, but then becomes impairing. More long data also consistently degrades the short-context performance.

## 3.3 CHOOSING A HIGH-QUALITY SHORT-CONTEXT MIX IS IMPORTANT

We saw in §2.3 that it is difficult to preserve the strong performance of Llama-3-8B on short-context tasks during long-context fine-tuning. We adopt our best long-context settings (Book/repo data and 60% long/40% short) and study the impact of different short-context training mixes. We experiment with SlimPajama (Soboleva et al., 2023), FineWeb-Edu (Penedo et al., 2024), DCLM-Baseline (Li et al., 2024a), and our own ProLong ShortMix. Our ShortMix is inspired by the "stage 2 training" in MiniCPM (Hu et al., 2024a) and Dolma-1.7 (Soldaini et al., 2024), which use more knowledge-intensive, downstream-related data at the end of pre-training. Table 5 shows the composition of our ShortMix.[3]

Table 5: Our ShortMix.

| Components | % |
|---|---|
| FineWeb | 25 |
| FineWeb-Edu | 25 |
| Wikipedia | 10 |
| Tulu-v2 | 10 |
| StackExchange | 10 |
| ArXiv | 10 |
| OpenWebMath | 10 |

---

[3]Since we do not truncate documents in the short data component unnecessarily, it includes a small percentage of documents longer than 8K tokens. See Table 14 in the appendix for the dataset length statistics.

Table 6: Impact of different short data sources. The long-context performance is the average of 6 categories at the lengths of 32K and 64K.

| Short Data (40%) | Long-Context | Short-Context | | | | | |
|---|---|---|---|---|---|---|---|
| | Avg. | HellaS. | MMLU | ARC-c | WG | GSM8K | Avg. |
| *Original model (Llama-3-8B)* | - | 82.1 | 66.5 | 59.4 | 77.1 | 44.7 | 66.0 |
| SlimPajama | 52.9 | 81.2 | 63.0 | 58.5 | 76.2 | 41.9 | 64.2 |
| FineWeb-Edu | 53.0 | 81.0 | 62.6 | 57.7 | 74.4 | 39.4 | 63.0 |
| DCLM-Baseline | 52.0 | 82.0 | 65.6 | 59.6 | 77.4 | 39.4 | 64.8 |
| ProLong ShortMix | **54.6** | 81.6 | 65.3 | 58.0 | 76.2 | 46.6 | **65.5** |

Table 6 demonstrates that the short data component has a substantial impact on both short-context and long-context downstream performance. Our curated ShortMix outperforms other short data sources on both short and long-context tasks and our data domains are particularly important for retaining Llama-3-8B's performance on mathematical reasoning. Surprisingly, we find that fine-tuning only using FineWeb-Edu—a dataset that is curated to help with knowledge-intensive tasks like MMLU—performs poorly as a short-context component, and we combine it with more diverse data sources in our ShortMix. DCLM-Baseline performs well on all short-context tasks except for GSM8K. This can likely be improved by combining with math-related datasets, but as we added the DCLM-baseline ablation at the conclusion of the project, we leave this exploration to future work.

## 4 SCALING THE SIZE AND LENGTH OF THE TRAINING DATA

Training for more steps is well-known to improve downstream tasks in regular pre-training, but little analysis has been done in the context of long-context continual training. We incorporate the lessons from our ablation experiments and arrive at the ProLong recipe, which we describe in detail in §6. Notably, we scale up the training budget to longer sequences (up to 512K) and more tokens (20B tokens at a maximum sequence length of 64K and an additional 20B tokens at 512K). We reset the learning rate schedule and increase the RoPE frequency base when switching from 64K to 512K context lengths. In this section, we analyze the impact of these decisions.

**Increasing the number of steps helps.** In Figure 4, we plot the downstream performance of intermediate checkpoints of our 40B-token runs. While the long-context performance fluctuates throughout training, we observe positive trends on recall, RAG, re-ranking, and summarization. For short-context tasks, we observe the average performance initially drops from the initialization, but gradually recovers. Performance again drops when switching from 64K to 512K sequence length, but also recovers with additional training.

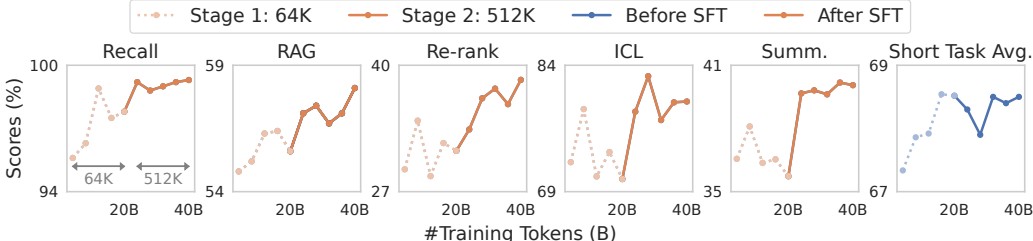

Figure 4: Performance of our final ProLong model throughout training.

**Increasing the training length beyond the evaluation length helps.** One might assume that we should train long-context models on the maximum sequence length that we want the model to support. Many works even emphasize the ability to extrapolate to even longer sequences at inference time (Press et al., 2022; Xiao et al., 2024b;a; Yen et al., 2024a; Chen et al., 2023). In contrast, we observe that training a model on an even longer sequence length (512K tokens) substantially improves the long-context performance at a shorter evaluation length (64K tokens).

Table 7: Impact of training models on different sequence lengths. All the results are evaluated at a sequence length of 64K. We see that training at a maximum length beyond the evaluation context window consistently improves the long-context performance.

| Max Seq. Length | Recall | RAG | Re-rank | ICL |
|---|---|---|---|---|
| ProLong 64K training (20B) | 96.5 | 52.7 | 22.8 | 70.6 |
| +4B 64K training | 95.0 | 56.4 | 28.0 | 78.8 |
| +4B 512K training | **98.5** | **56.9** | **32.9** | **79.2** |

We establish this by initializing with a model that was trained for 20B tokens at 64K and either (1) continuing training at 64K, or (2) switching to the 512K training. We use the same hyperparameters and data mixtures in either experiment. We evaluate a checkpoint after 4B training tokens on the recall, RAG, re-ranking, and ICL tasks at a evaluation length of 64K. Comparing the two runs in Table 7, we see consistent gains from switching to the 512K training length.[4]

# 5 SUPERVISED FINE-TUNING FOR LONG-CONTEXT LMS

In this section, we study how to best enable long-context language models to follow instructions. We focus on supervised fine-tuning on instruction datasets (Ouyang et al., 2022) and leave reinforcement learning and preference optimization for future work.

All our experiments in this section use the ProLong base model, which was trained for 40B tokens at a maximum sequence length of 512K. In comparison, open-source instruction data are very short, e.g., UltraChat (Ding et al., 2023) conversations have 1.2K tokens on average and 4.1K tokens maximum. To bridge this gap, several works (Xiong et al., 2023; Dubey et al., 2024; Xiong et al., 2024) have proposed to generate long-context instruction tuning data synthetically.

We consider three popular SFT datasets—UltraChat (Ding et al., 2023), Tulu-v2 (Ivison et al., 2023), ShareGPT[5]—and three sources of synthetic data: For *synthetic QA*, we prompt Llama-3-8B-Instruct to generate a question-and-answer pair given a random chunk from a long document; we reuse the QA pairs for *synthetic RAG* but we present a random list of chunks from the document to mimic retrieved passages; for *synthetic summarization*, we generate summaries for long books via recursive summarization (Wu et al., 2021). For all synthetic data, we write several templates, which we sample at random to increase diversity. More details can be found in §B.5. We always use a combination of 40% synthetic QA, 30% synthetic RAG, and 30% synthetic summarization in our synthetic instruction dataset. The hyperparameters for the instruction tuning experiments can be found in Table 9.

**Short-context instruction data yields strong long-context results.** We first establish that UltraChat outperforms Tulu-v2 and ShareGPT in Table 22. We therefore use it when studying the ratio of synthetic long-context instruction data in Table 8. Surprisingly, we find that adding synthetic data does not improve the performance on these very long-context tasks, and adding even as little as 1% synthetic data hurts the performance in our setting. Therefore, we use only short-context UltraChat data for the instruction tuning of our final ProLong model.

Table 8: Effect of different ratios of synthetic SFT data (mixed with UltraChat). We report the 32K-and-64K-averaged performance except tasks marked with $^\dagger$, which are evaluated at 512K for stress testing. The number of percentage is based on #tokens, not #samples.

| % Synthetic Data | JsonKV$^\dagger$ | RAG | Re-rank | ICL | QA$^\dagger$ | Summ.$^\dagger$ | Avg. |
|---|---|---|---|---|---|---|---|
| 0% | 65.7 | 58.1 | 38.5 | 80.3 | 49.7 | 42.1 | **55.7** |
| 1% | 61.5 | 57.0 | 38.3 | 80.8 | 45.3 | 41.5 | 54.1 |
| 3% | 62.0 | 56.4 | 37.9 | 80.6 | 44.8 | 39.5 | 53.5 |
| 10% | 70.3 | 55.5 | 36.1 | 80.6 | 41.7 | 39.4 | 53.9 |
| 50% | 45.8 | 48.8 | 18.8 | 70.5 | 42.3 | 33.3 | 43.3 |

---

[4]While we demonstrate the benefit of longer data, we note that training with longer sequences is more expensive, and may therefore not be the computationally optimal choice.

[5]https://huggingface.co/datasets/RyokoAI/ShareGPT52K.

Table 9: The training recipe for ProLong. Note that compared to our data ablations in §3, we decide to add textbooks data, which slightly changes the proportions of ShortMix.

| **Continued Long-context Training** | | |
|---|---|---|
| **Data** | 30% code repos, 30% books, 3% textbooks, 37% ShortMix | |
| | ShortMix: | 27% FineWeb-Edu, 27% FineWeb, 11% Tulu-v2, 11% StackExchange, 8% Wikipedia, 8% OpenWebMath, 8% ArXiv, |
| **Length Curriculum** | Stage 1 (64K): | Code repos, books, and textbooks at length 64K |
| | Stage 2 (512K): | Code repos: 50% at length 512K, 50% at length 64K Books: 17% at length 512K, 83% at length 64K Textbooks at length 512K |
| **Steps** | Stage 1: 20B tokens (2.2K H100 hours),   Stage 2: 20B tokens (12.2K H100 hours) | |
| **Model** | Initialization: RoPE: Attention: | Llama-3-8B-Instruct (original RoPE base freq. $5 \times 10^5$) Stage 1: $8 \times 10^6$, Stage 2:   $1.28 \times 10^8$ Full attention with cross-document attention masking |
| **Optim.** | AdamW (weight decay = 0.1, $\beta_1 = 0.9$, $\beta_2 = 0.95$) LR:                       $1e-5$ with 10% warmup and cosine decay to $1e-6$, each stage Batch size:         4M tokens for stage 1, 8M tokens for stage 2 | |
| **Supervised Fine-tuning (SFT)** | | |
| **Data** | UltraChat | |
| **Steps** | 1B tokens | |
| **Optim.** | AdamW (weight decay = 0.1, $\beta_1 = 0.9$, $\beta_2 = 0.95$) LR = $2e-5$ (cosine decay to $2e-6$), warmup = 5% Batch size = 4M tokens | |

Why do our conclusions about synthetic data differ from previous work? We offer the following hypotheses: (1) Previous work like Xiong et al. (2024); Bai et al. (2024a) may have insufficient long-context training and the synthetic data acts as additional long-context training data. (2) Our instruction dataset is much smaller compared to the private instruction data used for Llama-3.1 (Dubey et al., 2024)—it is possible that when using an extensive short instruction dataset, mixing in synthetic long data avoids the model from degenerating on long-context tasks.

# 6   THE PROLONG MODEL: RECIPE AND RESULTS

## 6.1   FINAL RECIPE

We summarize the training recipe for ProLong in Table 9. Our final model starts from the Llama-3-8B-Instruct model and is trained on 64K sequence length for 20B tokens. It is then further trained on 512K sequence length for 20B tokens (ProLong base), which we achieve using sequence parallelism (Li et al., 2023). We obtain the final ProLong model via SFT of the base model on UltraChat. One small difference on the data mixture between our ablations and the final model is that we mix in 3% high-quality textbooks (Chevalier et al., 2024), as book-like data are shown to be beneficial for long-context (§3.1) and textbooks are also highly educational. You can find more details about our data processing (§B.2) and the training stack (§B.3) in the appendix.

In the following, we elaborate on several carefully ablated design choices in our recipe.

**RoPE frequency base tuning.** We find that changing the RoPE (Su et al., 2021) frequency base to achieve position extrapolation (Xiong et al., 2023; emozilla, 2023) significantly improves long-context performance, even with a significant amount of training. §C.1 shows our ablation on the best RoPE base to use. While the original Llama models use a RoPE base of $10^5$, we use a base of $8 \times 10^6$ for the 64K setting and $1.28 \times 10^8$ for the 512K setting.

**Disabling cross-document attention.** Ding et al. (2024a) show that masking out attention across document boundaries improve model performance and this was also used during Llama-3 pre-training

Table 10: Our main evaluation results on HELMET (Yen et al., 2024b; details in §B.1). All results are averaged over sequence lengths of 32K, 64K, and 128K. For all models, we use the corresponding instruction version. ProLong is one of the best performing 10B-scale LMs while achieving a maximum context window of 512K tokens. The complete set of results can be found in §D.

| Model | Max Len. | Recall | RAG | ICL | Re-rank | QA | Summ. | Avg. |
|---|---|---|---|---|---|---|---|---|
| ProLong (8B) | 512K | **99.4** | **66.0** | 81.1 | **33.2** | 40.8 | 40.5 | **60.2** |
| MegaBeam-Mistral (7B) | 512K | **99.4** | 58.1 | **82.1** | 22.1 | 33.7 | 43.6 | 56.5 |
| Meta-Llama-3.1 (8B) | 128K | 98.7 | 62.8 | 79.7 | 26.6 | 40.4 | **46.1** | 59.0 |
| Qwen2 (7B) | 128K | 34.4 | 43.4 | 54.8 | 4.6 | 23.3 | 38.5 | 33.2 |
| Phi-3-small (7B) | 128K | 74.8 | 60.6 | 82.0 | 18.5 | 34.1 | 42.4 | 52.1 |
| Mistral-Nemo (12B) | 128K | 24.9 | 48.1 | 82.0 | 4.7 | 37.7 | 37.0 | 39.1 |
| Jamba-1.5-Mini (12B/52B) | 256K | 87.7 | 61.3 | 88.4 | 25.9 | 42.0 | 38.6 | 57.3 |
| Meta-Llama-3.1 (70B) | 128K | 98.5 | 65.9 | 80.0 | 39.4 | 47.2 | 51.1 | 63.7 |
| Claude-3.5-Sonnet | 200K | 99.4 | 44.0 | 79.3 | 19.9 | 38.1 | 49.2 | 55.0 |
| Gemini-1.5-Pro | 2M | 94.2 | 71.4 | 78.9 | 65.3 | 44.4 | 56.2 | 68.4 |
| GPT-4o | 128K | 99.9 | 71.5 | 86.7 | 59.6 | 47.0 | 55.7 | 70.1 |

(Dubey et al., 2024). In §C.2, we show that disabling cross-document attention in continual training benefits both the short and long-context performance. Disabling cross-document attention can also result in higher training throughput, which we describe in more detail in §B.3.

**Starting from Llama-3-8B-Instruct.** While we conduct all our long-context training ablations with the base model of Llama-3-8B, we use Llama-3-8B-Instruct as the initialization for the final ProLong model. §C.3 shows that while slightly improving the long-context performance, Llama-3-8B-Instruct significantly enhances the short-context performance.

## 6.2 PROLONG PERFORMANCE

We present the final HELMET evaluation results of ProLong in Table 10. We compare to a number of frontier long-context LMs, namely MegaBeam[6], Llama-3.1 (Dubey et al., 2024), Qwen2 (Yang et al., 2024a), Phi-3 (Abdin et al., 2024), Mistral-Nemo[7], Jamba-1.5 (Lenz et al., 2024), Claude-3.5-Sonnet (Anthropic, 2024), Gemini-1.5 (Reid et al., 2024), and GPT-4o (Achiam et al., 2023).

ProLong outperforms all 10B-scale models on our long-context evaluation. Notably, ProLong outperforms Llama-3.1-8B-Instruct on all long-context categories except summarization. ProLong achieves this with only 5% of Llama-3.1's long-context training data budget (40B vs. 800B tokens).

Since most existing models do not support more than 128K tokens, to showcase ProLong's 512K context length, we stress test ProLong on the QA and summarization tasks from 32K to 512K[8]. Table 11 shows that the performance of ProLong continues to improve at longer lengths, suggesting an effective longer context window.

Table 11: ProLong at 512K.

| | 32K | 64K | 128K | 512K |
|---|---|---|---|---|
| QA | 31.7 | 43.7 | 46.7 | **49.7** |
| Summ | 40.4 | 39.8 | 41.5 | **42.1** |

Besides HELMET, we also evaluate our models on NoCha (Karpinska et al., 2024)—a claim verification dataset on 67 recently published English fictional books. We chose this dataset because (1) it minimizes the data contamination problem as all the books are unlikely to exist in the model pre-training data; (2) all the claims are written by human readers and require global reasoning. Each test instance contains two contradictory claims, and the models must correctly judge both to pass.

Table 12 demonstrates the NoCha evaluation results. Among 10B-scale models, ProLong achieves the best accuracy on the extremely long test instances (>180K); on test instances <75K tokens, ProLong significantly outperforms other models and is the only model that is better than random guessing (25%). This further showcases the strength of our training recipe and the ProLong model.

---

[6] https://huggingface.co/aws-prototyping/MegaBeam-Mistral-7B-512k.

[7] https://huggingface.co/mistralai/Mistral-Nemo-Instruct-2407.

[8] In QA and summarization, we truncate the documents at the evaluation length; hence an effective long-context model should demonstrate better performance on longer lengths.

[9] https://github.com/marzenakrp/nocha. NoCha has a private test set and all evaluation is done by the NoCha authors. Hence, we report models from Table 10 that are also evaluated by the NoCha leaderboard.

Table 12: Results on the NoCha benchmark (Karpinska et al., 2024).[9] ProLong is the only model that achieves above-random performance in the <75K category and we consistently beat Llama-3.1. Different from the original NoCha leaderboard, we report the average accuracy over all test instances without filtering the test examples based on the model's context window lengths.

| Model | Max Len. | <75K | 75K-127K | 127K-180K | >180K |
|---|---|---|---|---|---|
| ProLong (8B) | 512K | **28.4** | 17.0 | 13.1 | **20.3** |
| MegaBeam-Mistral (7B) | 512K | 19.8 | **18.3** | **17.5** | 15.6 |
| Meta-Llama-3.1 (8B) | 128K | 17.3 | 16.4 | 0.0 | 0.0 |
| Mistral-Nemo (12B) | 128K | 13.6 | 0.4 | 0.0 | 0.0 |
| Jamba-1.5-Mini (12B/52B) | 256K | 27.2 | 28.0 | 24.4 | 6.2 |
| Meta-Llama-3.1 (70B) | 128K | 42.0 | 25.0 | 0.0 | 0.0 |
| Gemini-1.5-Pro | 2M | 24.7 | 38.8 | 35.3 | 46.9 |
| GPT-4o | 128K | 55.6 | 58.4 | 0.0 | 0.0 |

## 7 RELATED WORK

**Adapting existing LMs for long contexts.** Many works explore extending the LM context windows with minimal training, either by position extrapolation (Chen et al., 2023; Peng et al., 2024; Chen et al., 2024; Ding et al., 2024b; Liu et al., 2024a; Zhang et al., 2024b; Zhu et al., 2024; Zhao et al., 2024; Wu et al., 2024; Hu et al., 2024b) or manipulating the attention patterns (Chen et al., 2024; Xiao et al., 2024b;a; Bertsch et al., 2023; Jin et al., 2024). Yoshida et al. (2020); Choromanski et al. (2021); Chevalier et al. (2023) instead explore the idea of compressing the long contexts into shorter forms. However, Fu et al. (2024); Lu et al. (2024) show that using full attention, applying simple position extrapolation, and fine-tuning the model on long documents reach much stronger results.

Llama 3.1 (Dubey et al., 2024) and Jamba (Lieber et al., 2024) achieve long-context capabilities by adding a long-context continued training stage between standard pre-training and supervised fine-tuning, which is the setting we follow. Fu et al. (2024) study the data engineering for this setting and argue that 0.5B tokens of domain-balanced, length-upsampled data is sufficient for acquiring the long-context recall ability—which we show is not sufficient if a more holistic evaluation is taken. Xiong et al. (2023); Dubey et al. (2024); Lieber et al. (2024); Xiong et al. (2024); An et al. (2024b); Bai et al. (2024a) also adopt synthetically-generated long data in the SFT stage; however, we find that using standard, short-context instruction data achieves the best long-context results in our setting.

**Efficient long-context architectures.** There have been many efforts in designing more efficient architectures, for example, linear attention/RNNs (Gu & Dao, 2023; Dao & Gu, 2024; Ma et al., 2022; Sun et al., 2023; Peng et al., 2023; Yang et al., 2024b), and alternative attention architectures (Rubin & Berant, 2023; Sun et al., 2024; Yen et al., 2024a). However, they often require training from scratch and many have the inherent limitations in terms of long-context recall (Jelassi et al., 2024; Arora et al., 2024). Recent works explore hybrid models (Waleffe et al., 2024; Lieber et al., 2024)) or distilling existing LMs into hybrid models (Wang et al., 2024) and show promising results.

**Long-context evaluation.** Many benchmarks have been proposed for long-context evaluation (Shaham et al., 2023; Hsieh et al., 2024; Krishna et al., 2023; Zhang et al., 2024a; An et al., 2024a; Bai et al., 2024b) There are works studying particular aspects of long-context LMs as well, such as positional bias (Liu et al., 2024b), in-context learning (Bertsch et al., 2024; Li et al., 2024b), and book-length summarization (Kim et al., 2024). In this work, we follow Yen et al. (2024b) for its diverse application coverage and reliable evaluations.

## 8 CONCLUSION

We study the problem of given a short-context pre-trained LM, how to most effectively continually pre-train and SFT the model to be long-context. We conduct thorough ablations on each component and many of our findings contradict existing practices or beliefs. We use all the findings to produce ProLong, a new state-of-the-art long-context LM. We release all our code, data, and models publicly and hope that our findings will boost research and applications of long-context LMs.

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

## A    LIMITATIONS

Although we try to ablate the major components of our training recipe, due to resource limitations, we cannot exhaust all aspects, such as the optimization hyperparameters and additional data mixtures. We also limit ourselves to the 10B-scale regime and the Llama-3 base models, which may limit the generalizability of our recipe to our base models. Another concern is that we are overfitting to the tasks chosen for model development—however, we do not directly train on those datasets and guiding model development with benchmark tasks has become a common practice in pre-trained LM development. We also show that our final recipe and model perform well on an additional evaluation dataset, NoCha.

## B    EXPERIMENT DETAILS

### B.1    EVALUATION

Table 13: The details for our long-context evaluation following HELMET (Yen et al., 2024b).

| Category | Metrics | Tasks and Datasets |
|---|---|---|
| **Recall** | SubEM | Given a randomly-generated long JSON file and a key, retrieve the corresponding value (Liu et al., 2024b). |
| **RAG** | SubEM | Given a question and many retrieved Wikipedia documents (shuffled), answer the question (Liu et al., 2024b). Datasets: *NaturalQuestion* (Kwiatkowski et al., 2019), *HotpotQA* (Yang et al., 2018), and *PopQA* (Mallen et al., 2023). |
| **Re-rank** | nDCG@10 | Given a query and many retrieved documents (shuffled), re-rank the top-10 documents. Datasets: *MSMARCO* (Bajaj et al., 2016). |
| **ICL** | Accuracy | Datasets selected from Bertsch et al. (2024): *TREC coarse*, *TREC fine* (Hovy et al., 2001), *NLU* (Liu et al., 2021), *Banking77* (Casanueva et al., 2020), and *Clinc-150* (Larson et al., 2019). |
| **QA** | GPT-4o score | Given a book, answer the question. Datasets (# tokens): *NarrativeQA* (medium: 73K; max: 518K; Kočiský et al., 2018). |
| **Summ.** | GPT-4o score | Summarize a given legal document. Datasets (# tokens): *Multi-LexSum* (medium: 90K; max: 5M; Shen et al., 2022) |

Table 13 shows all the datasets we used for the long-context evaluation from HELMET (Yen et al., 2024b). We highlight some of the evaluation protocol improvements that HELMET implemented compared to previous benchmarks here:

- **Sufficient context lengths and fine-grained control**. HELMET can evaluate models at a context length of 128K tokens and beyond. The evaluation protocol also allows for reporting results at different lengths, giving developers fine-trained controls for different needs of long contexts.

- **Better synthetic recall tasks**. As shown in HELMET, needle-in-a-haystack (Kamradt, 2024) is mostly saturated because of its simplicity—the model only needs to find a needle in some irrelevant context. We instead use the more challenging JSON KV task, first proposed in Liu et al. (2024b) and included in HELMET, where the model is required to find the corresponding value to a given key among a large JSON file.

- **Using class-balanced demonstrations and abstract labels for ICL**. To disentangle models' ability of learning from demonstrations from their pre-training bias of the task or the dataset label distribution (Pan et al., 2023), HELMET samples the same number of demonstrations for each class and uses number labels (*1*, *2*, ...) instead of natural-language labels (e.g., *location*, *description*, ...).

- **Model-based evaluation for long-context QA and summarization**. Instead of using traditional metrics like ROUGE (which has shown to be poorly indicative of the real model performance: Deutsch & Roth, 2021; Deutsch et al., 2022; Goyal et al., 2023; Chang et al., 2024), HELMET uses model-based evaluations to compare the reference answer and the model output. For QA,

HELMET uses GPT-4o to score the model output given the question and the reference answer at a 0-3 scale. For summarization, HELMET takes a similar approach as Zhang & Bansal (2021); Gao et al. (2023): it first uses GPT-4o to decompose the reference summary and the model summary into atomic claims; then it uses GPT-4o to check whether each reference atomic claim is covered by the model output (recall) and whether each model atomic claim is covered by the reference summary (precision). Yen et al. (2024b) show that the model-based evaluation correlates with human perceptions significantly better than traditional metrics.

## B.2 DATA PROCESSING

**Data sources.** We list all the data sources we have explored in our ablations and main experiments here: the Stack (Kocetkov et al., 2023), SlimPajama (Together, 2023; Soboleva et al., 2023), FineWeb (we use the 2023-50 snapshot), FineWeb-Edu (we use a random sample) (Penedo et al., 2024), Tulu-v2 (Ivison et al., 2023), OpenWebMath (Paster et al., 2024), textbooks (Chevalier et al., 2024), and Dolma (Soldaini et al., 2024). The Books, StackExchange, and ArXiv data are from SlimPajama. The Wikipedia data are from Dolma.

**Data filtering and packing.** For the short training data and the SFT data, we randomly sample and concatenate the documents or conversations into 64K chunks. The last document for each chunk is truncated. The truncated part is used as the beginning for the next chunk for the short training data but is discarded for the SFT data. For the long-context training data, we filter out the documents that are shorter than 64K; we do the same for the 512K setting.

**Final data mixture.** We use a slightly different long data mixture in our ablations (Table 5) and our main ProLong experiment (Table 9). For the final model, we mix 3% textbooks into the long-context training data. The textbooks are open-source resources from libretexts.org, collected and made available by Chevalier et al. (2024). We pre-process the data by concatenating chapters from the same text books, as well as books from the same subject areas. This results in extremely long sequences which we pack into contexts of either 64K or 512K tokens. Though we do not have an ablation for adding this data due to limited resources, we believe that it should have a slight positive effect to the final model performance as textbooks are highly educational long-context data.

Table 14: % Proportion of long documents for the short data components used in Table 6. These statistics are computed after packing and truncation and therefore correspond to the document lengths as seen by the model. We highlight that the proportion of documents beyond 32K is below 1% for ShortMix.

|  | >4K | >8K | >16K | >32K |
| --- | --- | --- | --- | --- |
| FineWeb | 1.4 | 0.3 | 0.1 | 0.0 |
| FineWeb-Edu | 2.8 | 0.8 | 0.2 | 0.0 |
| Wikipedia | 1.6 | 0.4 | 0.0 | 0.0 |
| Tulu-v2 | 0.0 | 0.0 | 0.0 | 0.0 |
| StackExchange | 0.6 | 0.1 | 0.0 | 0.0 |
| ArXiv | 85.7 | 64.0 | 30.3 | 7.6 |
| OpenWebMath | 11.1 | 4.3 | 1.2 | 0.3 |
| ShortMix | 10.9 | 7.2 | 3.2 | 0.8 |
| SlimPajama | 11.3 | 7.4 | 4.9 | 3.2 |
| FineWeb-Edu | 2.8 | 0.8 | 0.2 | 0.0 |
| DCLM-Baseline | 4.9 | 1.7 | 0.4 | 0.1 |

## B.3 IMPLEMENTATION DETAILS

**Technical stack.** We use various open-source packages and tools for the ProLong training and evaluation. We use PyTorch (Paszke et al., 2019) and Hugging Face transformers (Wolf et al., 2020) for the model training. We use mosaic-streaming (Mosaic ML, 2022) for loading and mixing the data

and FlashAttention 2 (Dao, 2024) for efficient attention implementation. We implement sequence parallelism based on DeepSpeed-Ulysses (Jacobs et al., 2023). For long-context evaluation, we use HELMET (Yen et al., 2024b) and for short-context evaluation, we use lm-eval-harness (Gao et al., 2021).

**Attention and batching.** Since we do document masking in attention §6, we use the variable-length attention implementation from FlashAttention 2 (Dao, 2024) to speed up long-context training: for sequences that are concatenations of multiple short documents, instead of computing the full attention with masking, we instead compute the attention for each individual document. Since the complexity of attention is quadratic to the sequence length, this improves the training speed. However, the improvement is negligible in a distributed training setting with FSDP, since GPUs processing short sequence batches have to wait on other GPUs processing long sequences. We therefore implement a smart batching algorithm: In our setting, a gradient step usually consists of multiple gradient accumulation steps, where each device processes a smaller minibatch. We sort all the minibatches per training step by the sum of the squared lengths of documents in the sequence. This leads to more balanced sequence lengths across the GPUs and effective speedups, as can be seen in Table 15, without affecting the gradient updates or loss during training. However, the efficiency gains are diminished when training with more GPUs, as this reduces the number of gradient accumulation steps.

Table 15: Throughput per device of our ablation runs from Table 20, when training with 8 Nvidia H100 GPUs with FSDP. Our strategy of reordering minibatches is important for realizing the speed benefits from variable-length attention.

|  | Throughput (tokens/s/GPU) |
| --- | --- |
| 64K full attention | 2770 |
| Variable-length attention | $2780_{(+0.4\%)}$ |
| + Minibatch reordering | $3095_{(+11.7\%)}$ |

**Token-averaged loss.** We found that in the SFT stage, the distribution of the training tokens (in SFT, the tokens from the instructions are masked out and the models are only trained on the responses) on each GPU device can be extremely imbalanced, especially when there is synthetic data (most tokens in a synthetic data instance are from the instruction). Conventional all-reduce loss in distributed training averages over the sequences instead of valid tokens, which skews the optimization and also our control over the domain proportions. Instead, we change the all-reduce loss to be the average over all valid training tokens. Bai et al. (2024a) implements their SFT loss in a similar way.

### B.4 THE ABLATION SETTING

For all our ablations, unless specified, we train the *base model* of Llama-3-8B (instead of Instruct) on a 64K sequence length for 5B tokens, with the same hyperparameters as specified in Table 9. We choose this context length, as it is the highest power of 2 value for which we can train without sequence parallelism. By default, we use the same training data as the 64K ProLong setting, except that we remove the textbooks and use the ShortMix proportions in Table 5. For SFT, we use the same settings as specified in Table 9.

### B.5 GENERATING SYNTHETIC SFT DATA

We prompt Llama-3-8B-Instruct to generate the synthetic data and Table 16 shows the prompt we used for generating the synthetic QA data for books. We also write predefined templates and randomly sample one for each synthetic instance to increase the diversity, and Table 17 provides some examples.

Table 16: Prompts for generating synthetic QA data.

```
Given the following snippet of a book, ask a relevant question and
provide the answer.  The question and the answer should follow the
following rules:

(1) The question should be specific enough that it can only be
answered with the snippet.  The question should also be interesting
and intellectual enough that a curious reader of the book would ask
about it.
(2) The question and the answer should be comprehensible given just
the whole book without highlighting the snippet.  With that being
said, the question should NOT refer to the snippet directly (e.g., do
NOT say things like "Question:  given the conversation in the snippet,
what ...").  The answer also should not mention "the snippet ..."
explicitly (assuming that the snippet is never provided), but it can
copy the snippet content as a reference when answering the question.
(3) The answer should be concise but also should provide references
to the book when needed.  For example, \Wellington Yueh betrayed the
Atreides, as the book mentioned, '...'".

*** Start of the snippet ***

{sampled snippet}

*** End of the snippet ***

Before generating the question and the answer, first reason about
what this snippet is about.  In your generation, stick to the
following format:

Reasoning:  this snippet is about ...
Question:  ...
Answer:  ...
```

Table 17: Examples for question prompts and templates used for generating diverse synthetic QA data. We sample one question prompt and one template each time and combine them with the documents and the generated QA pairs to form a synthetic training example.

| Example question prompts for synthetic QA data |
|---|
| ```
Given the document, please answer the question.
Here is a piece of text; answer the following question based on it.
Please answer the question using the provided content.
Based on the given passage, respond to the question.
Read the snippet and answer the question that follows.
Using the provided text, answer the following question.
``` |
| Example templates for combining questions, answers, and contexts for synthetic QA data |
| ```
{prompt}\n\n{documents}\n\nQuestion:  {question}
{prompt}\n\n==== document starts ====\n{documents}\n==== document ends
====\n\nQuestion:  {question}
{prompt}\n\n{documents}\n\n{question}
{prompt} Question:  {question}\n\n{documents}
{prompt} {question}\n\n{documents}
{prompt}\n\n{question}\n\n{documents}
``` |

# C MORE ABLATIONS

## C.1 POSITION EXTRAPOLATION

Xiong et al. (2023); emozilla (2023) show that changing the RoPE frequency base to a larger value in continual long-context pre-training or in inference time can improve the long-context performance. emozilla (2023) suggests that one should scale the frequency base by a factor of $t^{\frac{d}{d-2}}$, where $t$ is the ratio between the target sequence length and the original LM length, and $d$ is the attention head dimension.

We conduct ablation studies, at both 64K (same as our standard ablation setting as specified in §B.4) and 512K (starting from ProLong-64K and training with the 512K data mixture for 5B tokens) sequence lengths, on what frequency bases we should use. Table 18 and Table 19 show the results. We first see that using the original 500,000 frequency base from Llama-3 leads to significant performance degradation. While dynamic NTK suggests $4 \times 10^6$, we find that further scaling it to $8 \times 10^6$ leads to better performance. Similar, we see that when scaling the 64K model to 512K, while dynamic NTK suggests a $64 \times 10^6$ frequency base, much larger frequency bases ($128 \times 10^6$ and $256 \times 10^6$) lead to better performance. We use $8 \times 10^6$ for 64K and $128 \times 10^6$ for 512K for our final ProLong models.

Table 18: Ablation study on RoPE frequency base at a maximum training length of 64K. Dynamic NTK (emozilla, 2023) roughly suggests to use 4m as the frequency base.

| RoPE Base ($\times 10^6$) | Long-Context | | | | | | | Short-Context |
|---|---|---|---|---|---|---|---|---|
| | Recall | RAG | Re-rank | ICL | QA | Summ. | Avg. | Avg. |
| 0.5 | 25.8 | 37.0 | 4.4 | 73.8 | 17.5 | 16.3 | 29.1 | 65.0 |
| 4.0 | 81.3 | 47.8 | 18.2 | 76.5 | 31.8 | 36.3 | 48.7 | 65.3 |
| 8.0 | 96.0 | 54.9 | 29.4 | 73.9 | 35.7 | 37.9 | **54.6** | **65.5** |

Table 19: Ablation study on RoPE frequency base at a maximum training length of 512K. Dynamic NTK (emozilla, 2023) roughly suggests to use $64 \times 10^6$ as the frequency base.

| RoPE Base ($\times 10^6$) | Long-Context | | | | | | | Short-Context |
|---|---|---|---|---|---|---|---|---|
| | Recall | RAG | Re-rank | ICL | QA | Summ. | Avg. | Avg. |
| 64 | 98.8 | 57.8 | 30.4 | 82.2 | 38.2 | 38.3 | 57.6 | 68.3 |
| 128 | 98.8 | 57.4 | 30.7 | 80.0 | 40.4 | 38.8 | 57.7 | **68.6** |
| 256 | 98.8 | 56.8 | 33.8 | 79.8 | 37.9 | 39.7 | **57.8** | 68.4 |

## C.2 DOCUMENT MASKS

We experiment whether to use document masks in attention in Table 20. Standard training concatenates multiple short documents into a single sequence (in our case, a 64K sequence), uses a special token to separate documents, and performs full attention over the whole sequence. When the document masks are used, we do not allow the attention to cross the document boundaries. We find that using document masks in continual long-context training leads to both better long-context results and short-context performance. For all our other ablations and the main experiment, we use document masks.

Table 20: Impact of using document masks in attention.

| Attention | Long-Context | | | | | | | Short-Context |
|---|---|---|---|---|---|---|---|---|
| | Recall | RAG | Re-rank | ICL | QA | Summ. | Avg. | Avg. |
| No doc masks | 97.4 | 53.6 | 20.4 | 76.6 | 37.2 | 36.3 | 53.6 | 64.9 |
| Document masks | 96.0 | 54.9 | 29.4 | 73.9 | 35.7 | 37.9 | **54.6** | **65.5** |

## C.3 INITIALIZATION

We use the base model for Llama-3-8B as the initialization for all our ablations to make sure the findings are generalizable and are not confounded by the Llama instruction tuning. However, for our final ProLong model, we use Llama-3-8B-Instruct as the initialization to achieve the best performance. We see in Table 21 (using the ablation setting from §B.4) that using Llama-3-8B-Instruct as the initialization achieves slightly better long-context performance and much stronger short-context performance.

Table 21: Differences of using the base Llama-3-8B model vs. Llama-3-8B-Instruct.

| Base Model | Long-Context | Short-Context | | | | | |
|---|---|---|---|---|---|---|---|
| | Avg. | HellaS. | MMLU | ARC-c | WG | GSM8K | Avg. |
| Llama-3-8B-Base | 54.6 | 81.6 | 65.3 | 58.0 | 76.2 | 46.6 | 65.5 |
| Llama-3-8B-Instruct | **55.0** | 80.8 | 66.1 | 58.5 | 75.6 | 57.7 | **67.7** |

## C.4 INSTRUCTION-TUNING DATASETS

Initialized from the ProLong base model, we experiment with different public, short-context SFT datasets. All runs use the same SFT hyperparameters as specified in Table 9. Table 22 shows that using UltraChat leads to the best overall results. Note that this does not necessarily mean that UltraChat is the best SFT dataset for all base models or applications.

Table 22: Ablations on using different short-context SFT datasets. We report the 32K-and-64K-averaged performance except tasks marked with $^\dagger$, which are evaluated at 512K for stress testing.

| SFT Data | Long-Context | | | | | | |
|---|---|---|---|---|---|---|---|
| | Recall$^\dagger$ | RAG | Re-rank | ICL | QA$^\dagger$ | Summ.$^\dagger$ | Avg. |
| UltraChat | 65.7 | 58.1 | 38.5 | 80.3 | 49.7 | 42.1 | **55.7** |
| Tulu v2 | 61.5 | 45.4 | 25.1 | 81.8 | 40.4 | 40.3 | 49.1 |
| ShareGPT | 40.5 | 47.5 | 26.7 | 79.6 | 42.7 | 34.4 | 45.2 |

## D FULL EVALUATION RESULTS

We provide all the evaluation results, including specific numbers for each dataset and evaluation length, at https://shorturl.at/pULbx.

## E EXPERIMENTS FOR REBUTTAL

We show additional results in Table 23, 24, 25, 26, 27 to address the concerns raised by the reviewers.

Table 23: An example of our synthetic data. The generated question and answer are fluent and faithful to the original book. One can refer to the content of the letter here to check the answer's correctness.

| Prompt |
| --- |
| Using the provided passage, answer the following question. What kind of job application did Eudora Welty submit to The New Yorker in 1933?

{Letters of Note: Correspondence Deserving of a Wider Audience} |

| Response |
| --- |
| Eudora Welty submitted a job application to The New Yorker in 1933, offering to write reviews of picture galleries and 15¢ movies, and expressing her ability to drum up an opinion after voracious reading, with a willingness to work from daylight to dark and even draw like Mr. Thurber if needed. |

Table 24: Effect of different ratios of synthetic SFT data (mixed with UltraChat). We report the 32K-and-64K-averaged performance except tasks marked with $^\dagger$, which are evaluated at 512K for stress testing. The number of percentage is based on #tokens, not #samples. "(8B)" and "(70B)" indicate that the synthetic data are generated by Llama-3-8B-Instruct or Llama-3-70B-Instruct. Even though using synthetic data from a stronger model leads to slightly better performance than using a weaker model, only using short-context SFT data still achieves the best result in our setting.

| % Synthetic Data | JsonKV$^\dagger$ | RAG | Re-rank | ICL | QA$^\dagger$ | Summ.$^\dagger$ | Avg. |
| --- | --- | --- | --- | --- | --- | --- | --- |
| 0% | 65.7 | 58.1 | 38.5 | 80.3 | 49.7 | 42.1 | **55.7** |
| 1% (from 8B) | 61.5 | 57.0 | 38.3 | 80.8 | 45.3 | 41.5 | 54.1 |
| 1% (from 70B) | 64.7 | 57.3 | 37.4 | 78.4 | 47.0 | 40.8 | 54.2 |
| 3% (from 8B) | 62.0 | 56.4 | 37.9 | 80.6 | 44.8 | 39.5 | 53.5 |
| 3% (from 70B) | 65.7 | 57.4 | 38.0 | 80.1 | 48.7 | 42.5 | 55.4 |
| 10% (from 8B) | 70.3 | 55.5 | 36.1 | 80.6 | 41.7 | 39.4 | 53.9 |
| 10% (from 70B) | 66.3 | 57.0 | 33.4 | 81.2 | 45.3 | 38.4 | 53.6 |
| 50% (from 8B) | 45.8 | 48.8 | 18.8 | 70.5 | 42.3 | 33.3 | 43.3 |
| 50% (from 70B) | 55.8 | 53.9 | 23.5 | 74.1 | 50.7 | 39.9 | 49.7 |

Table 25: Comparison between Fu et al. (2024) and our model. For a fair comparison, we use the same initialization (Llama-3-8B), same amount of data (5B), and same hyperparameters (§B.4). The ProLong data mix significantly outperforms Fu et al. (2024) on both short and long-context tasks.

| Data | Long-Context (After SFT) | | | | | | | Short-Context (Avg.) | |
| --- | --- | --- | --- | --- | --- | --- | --- | --- | --- |
| | Recall | RAG | Re-rank | ICL | QA | Summ. | Avg. | Before SFT | After SFT |
| Fu et al. (2024) | 95.8 | 52.1 | 23.1 | 72.0 | 31.0 | 37.0 | 51.8 | 64.1 | 65.4 |
| Our data mix | **96.0** | **54.9** | **29.4** | **73.9** | **35.7** | **37.9** | **54.6** | **65.5** | **67.5** |

Table 26: Short-context performance of our model *after* SFT. We also report a baseline using Llama-3-8B as the initialization and data from Fu et al. (2024), trained with 5B tokens. ProLong is initialized from Llama-3-8B-Instruct. "Llama-3-8B-Instruct + UltraChat": for a more fair comparison to ProLong, we conduct SFT on top of Llama-3-8B-Instruct with UltraChat. ProLong largely retraines the short-context performance of Llama-3-8B-Instruct except MMLU and GSM8K. We hypothesize that the close-source instruction tuning data of Llama-3-8B-Instruct is heavily engineered to improve math and knowledge-intensive tasks, which we do not have access to. ProLong achieves comparable results to "Llama-3-8B-Instruct + UltraChat", which further demonstrates that our data mix effective retains short-context performance.

| Model | HellaSwag | MMLU | ARC-c | WinoGrande | GSM8K | Avg. |
|---|---|---|---|---|---|---|
| Llama-3-8B + Fu et al. (2024) | 82.5 | 63.9 | 63.6 | 75.1 | 42.2 | 65.4 |
| Llama-3-8B | 82.1 | 66.5 | 59.4 | 77.1 | 44.7 | 66.0 |
| Llama-3-8B-Instruct + UltraChat | 82.1 | 65.1 | 64.3 | 75.5 | 60.7 | 69.5 |
| ProLong | 82.8 | 64.6 | 64.7 | 76.2 | 58.9 | 69.4 |
| Llama-3-8B-Instruct | 78.5 | 67.0 | 60.8 | 74.2 | 68.5 | 69.8 |

Table 27: Evaluation on additional benchmarks. Here we report the results on RULER (Hsieh et al., 2024) and ∞Bench (Zhang et al., 2024a) at 128K. As pointed out by Yen et al. (2024b), RULER and ∞Bench cannot reliably reflect long-context performance as their domain coverage is narrow and their evaluation metrics are noisy—as a result, we see unintuitive trends such as Gemini-1.5-Pro and Llama-3.1 (70B) perform worse than Llama-3.1 (8B). Regardless, our model still achieves the best performance on ∞Bench among all 10B-scale models.

| Model | RULER Avg. | ∞Bench MC | QA | Sum | Diag | Calc | Find | Number | PassKey | KV | Avg. |
|---|---|---|---|---|---|---|---|---|---|---|---|
| ProLong (8B) | 71.9 | 65.1 | 22.0 | 19.8 | 4.5 | 0.0 | 27.4 | 100.0 | 100.0 | 92.8 | **48.0** |
| MegaBeam-Mistral | 78.9 | 53.7 | 18.5 | 24.8 | 12.0 | 0.0 | 24.3 | 99.7 | 100.0 | 36.4 | 41.0 |
| Meta-Llama-3.1 (8B) | 81.3 | 67.2 | 15.5 | 26.7 | 23.0 | 0.0 | 33.1 | 99.5 | 100.0 | 55.0 | 46.7 |
| Qwen2 | 26.7 | 39.7 | 5.2 | 15.5 | 8.5 | 0.0 | 24.9 | 76.3 | 94.6 | 0.0 | 29.4 |
| Phi-3-small | 72.6 | 71.6 | 8.4 | 24.0 | 20.0 | 0.0 | 31.7 | 100.0 | 100.0 | 19.6 | 41.7 |
| Mistral-Nemo | 22.7 | 31.9 | 16.8 | 14.3 | 5.5 | 0.0 | 1.4 | 36.6 | 62.7 | 0.0 | 18.8 |
| Jamba-1.5-Mini | 87.8 | 76.0 | 17.9 | 0.0 | 3.5 | 0.0 | 31.1 | 100.0 | 100.0 | 45.6 | 41.6 |
| Meta-Llama-3.1 (70B) | 75.8 | 75.5 | 23.3 | 31.3 | 18.0 | 0.0 | 43.1 | 99.7 | 100.0 | 2.6 | 43.7 |
| GPT-4o-mini | 80.8 | 78.2 | 19.1 | 24.8 | 21.5 | 0.0 | 69.7 | 100.0 | 100.0 | 80.4 | 54.9 |
| GPT-4o | 93.3 | 86.5 | 26.0 | 21.5 | 51.0 | 0.0 | 58.9 | 100.0 | 100.0 | 99.8 | 60.4 |
| Gemini-1.5-Pro | 65.3 | 77.5 | 27.7 | 29.0 | 97.5 | 0.0 | 58.0 | 100.0 | 100.0 | 70.4 | 62.2 |

