# OpenReview forum: "How to Train Long-Context Language Models (Effectively)"
_ICLR.cc/2025/Conference — Submitted to ICLR 2025_

### Official Review · Reviewer_GrHC · 2024-10-17

**Soundness:** 1
**Presentation:** 4
**Contribution:** 2
**Rating:** 6
**Confidence:** 4

**Summary:**

The paper addresses the challenges involved in training language models (LMs) to make effective use of long-context information. It proposes a systematic methodology that goes beyond traditional evaluation metrics, such as perplexity, to train models capable of handling long-context inputs more effectively. The authors introduce ProLong-8B, a model initialized from Llama-3 and trained on 40 billion tokens, demonstrating state-of-the-art performance for long-context tasks.

**Strengths:**

1.  The study highlights that training on a mix of both long and short-context data sources is key to achieving good performance across tasks. The authors show that using long-context data like code repositories and books, balanced with high-quality short-context data, preserves both long and short-context abilities of the model.
2.  The extensive ablation experiments provide meaningful insights into effective training practices. For instance, the paper reveals that training only on long-context data negatively impacts overall performance and that including high-quality short-context data is essential for optimal results.
3. The resulting ProLong-8B model achieves strong performance on long-context tasks compared to similarly sized models

**Weaknesses:**

1. **Misalignment Between Title and Content**: A significant portion of the paper focuses on data engineering rather than providing comprehensive training methodologies for long-context language models, which does not fully align with the promise of the title. To better align with the title, the paper could have included more discussions on training techniques, such as **pose: efficient context window extension of LLMs via positional skip-wise training** and **CLEX: Continuous Length Extrapolation for Large Language Models**. Additionally, incorporating experiments involving different rotary position encoding methods and synthetic data, such as **LongSkywork: A Training Recipe for Efficiently Extending Context Length in Large Language Models**, would provide a more thorough exploration of how to train long-context LLMs.

2. **Benchmark Justification**: In Section 2, the necessity of introducing a new benchmark is not clearly justified, especially considering that many of the tasks overlap with existing benchmarks like **RULER**. Providing more rationale for the benchmark's uniqueness？

3. **Lack of Explanation**: In Section 3.2, the authors note that previous techniques (using long text) deteriorate short-context performance, but they do not provide a satisfactory explanation for this phenomenon. Without a clear rationale, the motivation for mixing in short-context training data remains weak.

4. **Insufficient Explanation of Data Mixture Ratios**: The paper proposes two mixture ratios—one for combining long-context data sources (books and repositories) and another for mixing long and short-context data. While the experiments show the effectiveness of these ratios, there is no clear explanation of why these particular proportions work.

5. **Lack of Comparison with Short-Context Performance Task**: The final experiments lack a comparison with short-context performance tasks. Including such comparisons would provide a more holistic evaluation of the model's performance.

**Questions:**

The paper's contributions are not clearly defined in terms of their nature, like technical report？ If the paper aims to contribute as a methodological study on data engineering, the novelty seems insufficient, as it mainly provides data mixture ratios without in-depth justification or exploration. On the other hand, if it aims to be an experimental study, the lack of diverse experiments and limited comparisons with other training approaches weakens its claims of novelty.

---

> ### Author Response · Authors · 2024-11-18
> **Response to Reviewer GrHC**
>
> Thank you for your thorough review and for highlighting that our extensive experiments provide meaningful insights!
> We respond to your concerns below and—since some concerns were shared with other reviewers—in the general response above!
>
> > Misalignment Between Title and Content
>
> We believe that evaluation-guided model development, data engineering, and SFT are some of the most important aspects in long-context training, which we thoroughly analyse and discuss in our paper. The significant improvement of ProLong compared to previous methods (such as [1]) and the competitive performance of ProLong compared to close-source long-context models (such as Llama 3.1) validates the title's claim that our training is *effective* at producing strong long-context models.
>
> We don't claim in this paper that our training method is *efficient*, i.e. extending models to long context windows with limited GPU memory or few training examples, as is the major contribution of the *Pose*, *CLEX* and *LongSkywork* papers.
> However, we agree with the reviewer that the mentioned references are important efforts in this field and we will add them to our related work discussion.
>
> [1] Fu et al., 2024. Data Engineering for Scaling Language Models to 128K Context.
>
>
> > Rationale for HELMET's uniqueness
>
> We highlight that we are *not* introducing HELMET as a new benchmark in this paper, but we are making use of it as an *existing benchmark*.
> In the general response, we provide additional evaluation results on RULER and InfiniteBench, and also highlight why HELMET is a good benchmark for model development.
>
> > Lack of Explanations
>
> A mechanistic explanation for the performance of certain data sources would be far out of the scope for this paper and beyond our current methods in machine learning. However, our rationale for the importance of short data is the following: Regular pre-training datasets are incredibly diverse and known to induce a wide range of capabilities in models. Selecting only the subset of long-form documents introduces bias, as we cannot assume that this subset will have the same coverage over all types of knowledge and reasoning as regular pre-training data. Therefore, to align the distribution with the pre-training distribution and prevent catastrophic forgetting, mixing in high-quality short-context data improves performance.
>
> > Lack of Comparison with Short-Context Performance Task
>
> This is an important point and we have added the final model short-context performance to address this concern! Please see our general response for details.

---

> > ### Comment · Reviewer_GrHC · 2024-11-19
> >
> > W1: Regarding "Misalignment Between Title and Content," I note that your title is "How to Train Long-Context Language Models (Effectively)," and the first line of the abstract states, "We study the problem of adapting a language model (LM) to make effective use of long-context information." I respectfully disagree with your position; in my view, the title and abstract imply a broader study that includes considerations of the training methods themselves.
> >
> > W2: For "Rationale for HELMET's Uniqueness," I reviewed the related results but am still a bit unclear on certain points. For instance, you mentioned that "RULER and InfiniteBench have some unintuitive trends." Could you clarify if these trends are due to differences in tasks or evaluation metrics? Additionally, since there is some task overlap between RULER and InfiniteBench, do you observe any performance discrepancies there? You also mentioned that HELMET has advantages such as (1) comprehensive coverage of applications and (2) stable model-based metrics for QA and summarization tasks. However, I did not find specific details about what constitutes the "stable model-based metrics." Could you provide further elaboration on this?
> >
> >
> > W3: Lack of Explanations,I agree with your rationale for why short-context data is important, and I also believe the reasoning likely to be correct. Still, it would be even more compelling if there were concrete evidence supporting this hypothesis.
> >
> > Lastly, it seems you may have missed addressing W4 and Q1. I would appreciate if you could provide further comments on these as well.

---

> ### Author Response · Authors · 2024-11-20
> **2. Response to Reviewer GrHC**
>
> Thank you for your fast response and giving us the opportunity for discussion!
>
> > W1: Misalignment Between Title and Content
>
> We appreciate the concerns about our title. Unfortunately, we are not allowed to revise the title at the current stage; However, we have revised our abstract’s first sentence (highlighted in red) to make the setting clear: “*We study continued training and supervised fine-tuning (SFT) of a language model (LM) to make effective use of long-context information.*” This should help put our paper in the correct context.
>
> We now revisit the training methods suggested by the reviewer and their relation to our paper:
> - PoSE focuses on efficiently adapting models to longer sequences by training with position skipping. In Table 1 of the PoSE paper, it underperforms training on long-sequence data directly (2nd row), which is our setting
> - CLEX proposes a novel position extrapolation. However, we point to a recent study [1], which shows that changing the RoPE frequency base (as used in our paper) performs better than CLEX and other methods, e.g., YARN.
> - LongSkywork proposes a method for chunking-and-shuffling pre-training documents, and a type of tabular synthetic instruction data. We agree that these would provide two additional interesting data points in our paper. We will work to add these in the final version of the paper, although it is difficult to build on LongSkywork, since no artifacts (data, models, code) were released.
>
> Overall, we still believe that our paper constitutes a broad study of the major ingredients for language model development: designing an evaluation framework, studying the pre-training data, and performing instruction tuning, while also covering details such as tuning the rope frequency base and disabling cross-document attention (Appendix C). It is perhaps helpful to say that we draw inspiration from impactful full-stack language model development papers such as Mini-CPM [2]  and OLMo [3].  Overall, we hope that the reviewer can approach our title with an open mind and focus on our contributions.
>
> > W2: Rationale for HELMET's Uniqueness
>
> We appreciate the insightful questions on evaluation and address them below. We also encourage the reviewer to refer to the original HELMET paper [4] for more details.
>
> - *Unintuitive trends on RULER and InfiniteBench*: HELMET observes that models are often penalized for generating ill-defined formats on RULER and InfiniteBench, and adds better prompts and in-context demonstrations. Additionally, InfiniteBench uses ROUGE as metrics for QA and summarizations. Figure 2 in the HELMET paper shows that HELMET's model-based evaluations  reveal clear distinctions between models where ROUGE scores are similar.
> - *Task overlap*: InfiniteBench and RULER both have synthetic “needle-in-a-haystack” recall tasks, but due to different prompt templates and “haystack” filler content, the performance can vary drastically for models. These differences lead to varying trends, though both benchmarks display unintuitive rankings.
> - *Model-based metrics*: In Appendix B.1, we discuss the reference-based GPT-4o evaluation metrics used in HELMET. The HELMET paper has more details, including experiments in Figure 2 and human evaluations in Appendix B.6, to show why model-based metrics are more accurate than traditional metrics like ROUGE in assessing model generations.
>
> > W3: Lack of explanations of using short-context data
>
> We appreciate that the reviewer found our explanation helpful! We can quantify this effect by directly measuring the diversity of the pre-training corpus via the SemanticDiversity metric proposed by [5]. We compute Semantic Diversity using embeddings from `Alibaba-NLP/gte-base-en-v1.5`  on a balanced set of 20K long and short documents from CommonCrawl, one of the most diverse data sources available. The long documents (>8192 tokens) attain a 50% lower diversity score than short documents (90 vs. 176). We note that during long-context training, this is compounded by one long document taking up as many tokens in a batch as dozens of short documents.
>
> While this supports our argument, the observed impact of the data on downstream performance remains an empirical finding. We would greatly value suggestions from the reviewer on what form of “concrete evidence” would be appropriate or examples from similar data engineering papers that could guide us.
>
> > Response W4 and Q1
>
> *See separate comment due to character limit*
>
> [1] Lu et al., 2024. A controlled study on long context extension and generalization in llms
>
> [2] Hu et al., COLM 2024. MiniCPM: Unveiling the Potential of Small Language Models with Scalable Training Strategies
>
> [3] Groeneveld et al., ACL 2024. OLMo: Accelerating the Science of Language Models
>
> [4] Yen et al., 2024. HELMET: How to Evaluate Long-Context Language Models Effectively and Thoroughly
>
> [5] Li et al., EMNLP 2024. ScalingFilter: Assessing Data Quality through Inverse Utilization of Scaling Laws

---

> ### Author Response · Authors · 2024-11-20
> **2. Response to Reviewer GrHC (continued)**
>
> *We also thank the reviewer for pointing out missing responses. We address them below.*
>
> > W4: Insufficient Explanation of Data Mixture Ratios
>
> We do not claim that those particular ratios would work universally but instead argue that these ratios can have a significant impact on the model performance and should be tuned carefully.
>
> For the book vs. code ratio, we found that while both domains alone perform well as long-context sources, each excels in specific tasks. We then took a heuristic approach to evenly combine them in equal parts and saw further improvements beyond either domain alone. We were not able to test more candidates due to resource constraints.
>
> For short vs. long, we studied the optimal ratio via a systematic grid search (Figure 3). We suspect that a simple, human-interpretable explanation for this particular value may be elusive. As an analogy, if we assume that short data acts as a form of “regularization” to prevent catastrophic forgetting of general-domain knowledge, then tuning the short-to-long data ratio is similar to a hyperparameter search of a regularization constant, which is common practice in machine learning. We appreciate the reviewer’s curiosity and would value any examples or insights that could help us make progress on these questions.
>
> > Q1: The paper's contributions are not clearly defined in terms of their nature, like technical report？
>
> We respectfully disagree with the reviewer’s assessment on our contribution. This paper focuses on **continual training and instruction tuning of an existing language model (LM) for long-context capabilities**. It is the first comprehensive open study of the entire pipeline, providing exhaustive ablations on key components, including evaluations, data engineering, scaling (training and sequence lengths), and SFT.
>
> Our work can be seen as a follow-up to Fu et al. [6], which also focused on long-context continued training. Besides improving the data engineering effort (see Table 25 in our paper), we significantly expanded the scope by studying the entire long-context training pipeline, including evaluation, scaling, and SFT. We designed careful experiments, which allow us to draw conclusions that challenge common wisdom in the long-context community, such as (1) some scaling trends in task performance *only emerge after SFT*, (2) there are benefits to training on *even longer sequences than used for evaluation*, and (3) *short SFT is effective at boosting long-context abilities*, after extensive long-context pre-training.
>
> Furthermore, we evaluated ProLong against the industry’s best-performing long-context models—which most long-context papers avoid—and showed that ProLong outperforms the state-of-the-art Llama 3.1 despite using only 5% of its long-context token budget.
>
> [6] Fu et al., 2024. ICML. Data Engineering for Scaling Language Models to 128K Context.

---

> > ### Comment · Reviewer_GrHC · 2024-11-22
> >
> > Thank you for your response. Assuming that our discussion and all explanations will be incorporated into the paper as much as possible, I have raised my score.

---

### Official Review · Reviewer_WrRx · 2024-10-17

**Soundness:** 3
**Presentation:** 4
**Contribution:** 4
**Rating:** 6
**Confidence:** 4

**Summary:**

This paper study the problem of how to effectively adapting a short-context language models to be long-context. The paper first starts with identifying deficiency in widely used perplexity and Needle-In-A-Heystack test and establishing a reliable evaluation protocol for long-context LLM by adopting HELMET, which covers diverse range of realistic long-context applications.
The authors also advocate for (1) checking model performance after supervised finetuning (2) ensuring that model performance on short-context tasks is not compromised after long context training. For long context data curation, the authors provide best practices for (1) data mixture in long-context data (2) quantity ratio between long and short context data (3) choosing high-quality short-context data.
Next, the paper discuss the impact of scaling training tokens and context length on both long and short task performance. Finally, the recipe ends with exploration on the choice of SFT data, pointing out that short-context instruction data can yield strong long-context result. The resulting long-context model ProLong as well as associated code and data are open-sourced.

**Strengths:**

1. The paper is very well structured and written, and is easy to follow.
2. The paper thoroughly explored important components and provide useful observations for long-context language models development, including training recipe and evaluation protocol. From the evaluation perspective, the authors underscore the importance of using diverse and realistic long-context tasks rather than relying on conventional perplexity-based and needle-in-a-haystack-like benchmarks. The preservation of short-context performance is also highlighted. From the training perspective, the authors conducted experiments and found empirically effective choices of data mixture, data ratio, and data quality.
3. The ultimately trained model, ProLong, achieves strong long context performance at 10B parameter level.
4. The assets associated with this paper could serve as useful resources for the research community and are open-sourced.

**Weaknesses:**

1. The choice of long-context evaluation benchmark(HELMET) seems arbitrary. The authors should discuss the rationale behind selecting HELMET instead of other long-context benchmarks, such as LongBench, Infinite Bench, etc.
2. The training-free length extrapolation methods compared in the paper is limited. The experiments would be more solid if more advanced training-free extension methods are incorporated, e.g., SelfExtend and ChunkLLaMa.
3. There exists other community-tuned long-context LLMs initialized from LLaMa-3-8B-Instruct, e.g., gradientai/Llama-3-8B-Instruct-Gradient-1048k. As they claim their training process only consume 1.4B tokens, it would be more convincing to include such models as strong baselines.

**Questions:**

See Weakness above.

---

> ### Author Response · Authors · 2024-11-18
> **Response to Reviewer WrRx**
>
> Thank you for your thorough review and finding many strengths in our paper! We address your remaining concerns below.
>
> > Choice of HELMET seems arbitrary
>
> We add more results on RULER and InfiniteBench and discuss the motivations for choosing HELMET in our general response.
>
> > Comparing to more TrainingFree length extrapolation methods
>
> Thank you for mentioning some interesting recent works on training-free methods.
> Several recent studies [1, 2] suggest that continuing training models on long-context data substantially outperforms any training-free methods, including SelfExtend. This motivates our study of *training-intensive* methods, making a thorough review of training-free methods outside the scope of this work. In Section 2, we only feature a simple position extrapolation baseline to motivate holistic evaluations considering both long and short-context performance.
>
> [1] Fu et al., 2024. Data Engineering for Scaling Language Models to 128K Context.
>
> [2] Lu et al., 2024. A Controlled Study on Long Context Extension and Generalization in LLMs.
>
> > There exist other community-tuned long-context LLMs
>
> Thank you for highlighting the gradientai model! We show the evaluation result for gradientai/Llama-3-8B-Instruct-Gradient-4191k below. Its performance on both long-context and short-context tasks is much lower compared to ProLong.
>
> | Model         | JsonKV | RAG   | Rerank | ICL  | Short |
> |---------------|--------|-------|--------|------|-------|
> | The gradientai model     | 82.8   | 50.3  | 6.8    | 83.2 | 62.1  |
> | ProLong       | 99.3   | 58.1  | 38.5   | 80.3 | 69.4  |

---

> > ### Author Response · Authors · 2024-11-22
> > **Reminder for paper discussion**
> >
> > Dear reviewer,
> >
> > As we approach the end of the discussion period, we would greatly appreciate your input on the paper. We hope our responses and additional results address your concerns and welcome any further questions or suggestions.

---

> > > ### Comment · Reviewer_WrRx · 2024-11-25
> > >
> > > Thanks for the authors' response. I have no further questions and I will keep my positive score.

---

### Official Review · Reviewer_DH4H · 2024-11-03

**Soundness:** 2
**Presentation:** 4
**Contribution:** 3
**Rating:** 6
**Confidence:** 5

**Summary:**

This paper investigates the optimal recipe of training a long-context LLM from the data engineering perspective. The authors investigate the effects of data source, mixing proportion of long and short data, etc, and evaluate the trained models on realistic tasks beyond perplexity.

**Strengths:**

The paper gives a thorough investigation of data engineering for long context training across various aspects, covering long-short performance balance, SFT, and evaluation. This paper can potentially serve as a handbook for long context training. The paper is well-organized and easy to understand.

**Weaknesses:**

- The evaluation beyond ppl is good and there are some previous studies [1] proprose ppl is not proper for evaluating long-context performance. However, it is kind of confused if the evaluation can serve as a "contribution" of this paper? It seems the paper just adopt the HELMET benchmark (I don't think the decision is a contribution as most long-context works have adopted some realistic benchmarks like LongBench [2] and InfiniteBench [3]).

- The paper proposed short-context instruction tuning is sufficient for achieving good performance. The study is conducted by mixing partial long synthetic instruction data. The authors found that the incorporation of long synthetic data would degrade the long-context performance.  As our understanding in short-context scenario, data quality is (most) essential for SFT [4]. However, the synthetic long-context data (generated by LLaMA-3-8B) is probably of low quality. So I think this study only demonstrates that long synthetic data is not good enough for improving long-context performance. BTW, have you compared the long-context performance before and after SFT? LLaMA 3.1 technical report [5] proposed the long-context performance would degenerate with short instruction data only after SFT. I wonder if this point holds here.

- I believe summarization is a vital and basic long-context ability, however, there is a significant performance drop in the summarization task after ProLong's training (compared to naive LLaMA-3.1). Is this due to the lack of some long data types/sources in training data (books and code only)? The information in long books and codes may be distributed without summarizing. I feel like arxiv/wikipedia  (which you only use in shortmix) are good data sources for summarization, as they naively contain an abstract section. The authors can consider including more data sources with ablations (as that in Table 4).

- Would you consider reporting results on more benchmarks like RULER [6] and InfiniteBench [3]? HELMET is a good but new benchmark while some experiences (e.g., the degeneration of long-context performance when using short instruction data only.) are acquired from evaluation on some older benchmark. More results would help us align the findings.


[1] Can Perplexity Reflect Large Language Model’s Ability in Long Text Understanding?

[2] LongBench: A Bilingual, Multitask Benchmark for Long Context Understanding

[3] ∞Bench: Extending Long Context Evaluation Beyond 100K Tokens

[4] LIMA: Less Is More for Alignment

[5] The Llama 3 Herd of Models

[6] RULER: What's the Real Context Size of Your Long-Context Language Models?

**Questions:**

See weaknesses above.

---

> ### Author Response · Authors · 2024-11-18
> **Response to Reviewer DH4H**
>
> We are glad that you found our paper thorough, well-organized and easy to understand, and we hope that we can address your concerns below!
>
> > Can evaluation serve as a "contribution" of this paper?
>
> While other papers have reported their *final model performance* on LongBench and InfiniteBench, we believe that we are the first to conduct a principled long-context development guided by such a downstream benchmark.
> As part of this contribution, we reveal that (1) this results in different outcomes than considering only perplexity or JsonKV, as done by prior work; (2) evaluating after SFT demonstrates distinct trends compared to before SFT; (3) it is important to also evaluate on short tasks which is neglected by prior work.
> We believe that this is an important shift in perspective as how the community should *develop* long-context models, and our contributions include the experimental insights we gain from this perspective.
> Thank you for pointing us to [1], which we will add to the discussion in our paper.
>
> > Weak synthetic SFT data quality
>
> This is an excellent point! We have updated the results with a stronger data generator (Llama-3-70B) and added discussion in comparison with Llama 3.1. Please see our general response above for more details on the results and takeaways.
>
> >  Have you compared the long-context performance before and after SFT?
>
> We show that the takeaways from Section 2.2 hold even for the final ProLong model. We see that SFT with only short SFT data in fact significantly improves performance on RAG and re-ranking, while hurting long-context ICL performance slightly.
>
> |            | JsonKV | RAG   | Rerank | ICL   | Short Performance |
> |------------|--------|-------|--------|-------|-------------------|
> | ProLong (before SFT) | 99.5   | 51.3  | 29.8   | 82.4  | 68.5             |
> | ProLong (after SFT)  | 99.3   | 58.1  | 38.5   | 80.3  | 69.4             |
> _Evaluated at 32K and 64K. We omit QA and summarization since their performance is much worse before SFT, as shown in Figure 2._
>
> >  Performance drop in summarization
>
> First we clarify that our model is initialized from *Llama-3-8B-Instruct* (original context length: 8K), instead of *Llama-3.1-8B-Instruct*. Hence it is not fair to say that after ProLong training, the summarization performance dropped!
>
> Nevertheless, we acknowledge that Llama-3.1-8B-Instruct outperforms ProLong on summarization. As Llama-3.1's extensive long-context and post-training pipeline is private and proprietary, we can only speculate that it is due to Llama-3.1’s extensive data curation, including preference optimization. However, we argue that without access to Llama’s million-dollar-worth SFT data, ProLong still achieves impressive performance relative to Llama-3.1.
>
> > More benchmarks like RULER and InfiniteBench
>
> Great suggestion! We have added RULER and InfiniteBench to the submission. Please refer to our general response above for more details.

---

> > ### Comment · Reviewer_DH4H · 2024-11-19
> > **Response to Authors**
> >
> > Thanks for your quick responses.
> >
> > - Now I get that the evaluation contribution is from using practical tasks to assess the strategies of long-context LLMs development and it makes sense. It is kind of like long-context data engineering based on practical evaluation that further extends previous studies such as [1] based on PPL only. I hope the authors can make this point clearer in the takeaways to avoid misunderstandings.
> >
> > [1] Data Engineering for Scaling Language Models to 128K Context
> >
> > - The claim in the revised version about long instruction data is more moderate and reasonable. The new results in general response seem to further confirm that  the long synthetic data from 8B model is of low quality (more data from 8B contributes to more drops compared to that of 70B.) I guess the data from 70B may still be not good and these results may prompt the community to do more efforts on long-context alignment.
> >
> > - The summarization performance of ProLong on InfiniteBench is still significantly worse than that of other domains not only compared to LLaMA 3.1.  I believe the summarization capability cannot be well-integrated from currently limited data sources (code and books only) and I do suggest including Arxiv/Wiki data in the future work as they involve many human-annotated abstracts for long papers/docs.
> >
> > - Thanks for reporting more results. Can you further list your evaluation settings of RULER and Infinitebench? It seems that the LLaMA 3.1 8B's results are not consistent with the official ones (https://github.com/NVIDIA/RULER).
> >
> > Generally, the first and second concerns can be addressed with some revisions. The third one is my suggestion and can be left for future work. For the fourth one, although the performance on RULER is not promising, I believe the added evaluation results can provide intuitions for further improvements to produce long-context LLMs following the paper.
> >
> > One more discussion:
> >
> > - Out of the goal to produce long-context LLM after SFT, it might be better to report results on instruction-following benchmarks like AlpacaEval and MT-Bench. I guess there should be some "continual training taxes" that the instruction-following capabilities of ProLong might be worse compared to naive LLaMA 3 Instruct.

---

> > > ### Author Response · Authors · 2024-11-20
> > > **2. Response to DH4H**
> > >
> > > Thanks for the fast response!
> > >
> > > > Evaluation contribution
> > >
> > > Exactly! We are glad that the reviewer agrees with our contribution and we will revise the paper accordingly to make this point clearer!
> > >
> > > > Synthetic data
> > >
> > > We agree with the reviewer that finding better ways to generate long-context instruction tuning data is a promising research direction and  will release all our data and code to facilitate future study.
> > >
> > > > Summarization  performance and future exploration in data engineering
> > >
> > > Thank you for the good suggestion! We will explore using more domains for long context data engineering in the future. We also believe that Llama 3.1’s extensive post-training pipeline significantly contributed to its better summarization result.
> > >
> > > > Differences in RULER/InfiniteBench settings
> > >
> > > Thanks for pointing this out! While the original RULER/InfiniteBench uses each model’s own tokenizer for length control, we instead use the same tokenizer (Llama 2) to ensure a fair comparison across models—this ensures that every model sees the same amount of content. Though this leads to slightly higher absolute numbers in our results (as the Llama 2 tokenizer results in fewer words compared to Llama 3/GPT tokenizers), the relative trends across models should be unchanged.
> > >
> > > > Additional evaluation on AlpacaEval or MT-Bench
> > >
> > > Another good suggestion! We agree that it would be interesting to have additional results on these benchmarks and will add them in the revision.

---

> ### Comment · Reviewer_DH4H · 2024-11-20
> **Response to Authors**
>
> I believe the new results can support me in raising the rating to 6. Hope the authors can include the promised results in revised versions and explore the discussed questions in future works.

---

> > ### Author Response · Authors · 2024-11-25
> > **Additional results on using arxiv long-context data**
> >
> > Dear reviewer,
> >
> > We have some additional results to share with you. We took your suggestions and ran ablations on using arXiv as a long data source. We updated Table 4 and also share the results here. The new results (arxiv, and books/repos/arxiv 1:1:1) are highlighted in _italic_ font.
> >
> > As we can see, arXiv is not as good of a long-context source as books/code repos, and combing all three does not perform as well as just using books and code repos. We hope this new results provide more insights into what constitutes good long-context sources.
> >
> > | Long data (60%)                          | Recall | RAG  | Re-rank | ICL   | QA   | Summ  | Long avg.   | Short avg. |
> > |----------------------------------|---------|--------|---------|--------|--------|--------|--------|--------|
> > | CommonCrawl                      | 84.1    | 53.3   | 28.1    | 67.5   | 35.2   | 37.0   | 50.9   | 66.5   |
> > | _ArXiv_                            | 90.3    | 51.8   | 28.0    | 68.0   | 33.7   | 36.7   | 51.4   | 67.5   |
> > | Books                            | 94.9    | 53.9   | **30.7**| 72.2   | 33.2   | 37.7   | **53.8**| 65.5   |
> > | Code Repos                       | **99.2**| 53.8   | 29.0    | 61.2   | 34.7   | 36.2   | 52.3   | **65.9**|
> > | _Books/Repos/ArXiv 1:1:1_          | 98.3    | 53.9   | 29.4    | 66.9   | 35.5   | 35.5   | 53.3   | 66.9   |
> > | **Books/Repos 1:1**              | 96.0    | **54.9**| 29.4   | **73.9**| **35.7**| **37.9**| **54.6**| 65.5   |

---

### Official Review · Reviewer_kqM8 · 2024-11-04

**Soundness:** 3
**Presentation:** 3
**Contribution:** 3
**Rating:** 6
**Confidence:** 5

**Summary:**

The paper proposes a continual training recipe, including data curation and multi-stage context extension, that successfully equips the llama-3-8B-instruct model with long context ability up to 512K tokens.

**Strengths:**

- The model trained with the final recipe (ProLong) shows strong performance on long context benchmarks.

- The proposed data mixture can be a valuable contribution for the community.

**Weaknesses:**

- Missing crucial baselines: The author should at least include a very baseline that applies the recipe of Fu et al. (2024) on the llama-3-8B model with 40B tokens to provide a fair comparison between the ProLong recipe and the Fu et al. (2024) recipe on both short and long context tasks. The Table 10 and 12 is not very informative because of lacking both fair comparisons and the short-context results.

- Misleading evaluation: As in L174, the short-context performance is evaluated before SFT, while the long context performance is after SFT (As in L149). However, what people really care about is if the final model after SFT can have both good short and long-context performances.

- The paper claims counterintuitive conclusions but lacks in-depth analyses explaining why they happen. The author claims that training on longer than evaluation context can bring performance benefits (in L331), but it can be simply due to the fact of training on 4B more tokens, which can improve the instruction following ability because of the short data mixture. The author claims that the long synthetic SFT data does not improve performance but it can be simply because of using a weak 8B model for data generation. See more details in the Question section.

**Questions:**

- Why training on all long context data will harm the downstream long-context tasks? Is it because (1) the proposed ShortMix data actually contains lots of QA data from Fineweb-Edu and Stack Exchange which can improve instruction following (2) the curated long context data (code repos+books) does not have QA data, and training purely on it will harm the instruction following ability?
In Line 351, why use the llama-3-8B-instruct model to synthesize long context SFT data, instead of more powerful models such as lama-3-70B-instruct? I would assume an 8B model cannot accomplish these difficult tasks and can provide low-quality long-context SFT data.

- Could you also evaluate ProLong on some popular synthetic long-context benchmarks such as RULER? In this way, we can have a better understanding of the limitations of ProLong.

- The stage 2 training also doubled the batch size from 4M to 8M. Do you have an ablation study for it? Batch size ramp-up is believed to provide more accurate optimization trajectory and boost model convergences, therefore the performance gains from 512K length training may actually be from the batch size ramp-up instead of longer context.

- Could you also evaluate the final ProLong model on short-context tasks, and add comparisons with a llama3-8B-instruct model SFTed with UltraChat?

---

> ### Author Response · Authors · 2024-11-18
> **Response to Reviewer kqM8**
>
> Thank you for your thoughtful review and finding that our data mixture can be a valuable contribution for the community! We hope to address your concerns and answer your great questions.
>
> > Missing Fu et al., (2024) baseline
>
> We added a direct and fair comparison in Table 25 (also shown below), where we train on (1) Fu et al. data and (2) ProLong data mix with the same hyperparameters and the same amount of tokens (5B). Using ProLong data mix clearly outperforms Fu et al. on both long and short-context performance.
>
> | Data              | JsonKV | RAG   | Re-rank | ICL   | QA    | Summ. | Long Avg.  | Short Avg. (before SFT) | Short Avg. (after SFT) |
> |-------------------|--------|-------|---------|-------|-------|-------|-------|------------|-----------|
> | Fu et al. (2024)  | 95.8   | 52.1  | 23.1    | 72.0  | 31.0  | 37.0  | 51.8  | 64.1       | 65.4      |
> | *Our data mix*    | **96.0** | **54.9** | **29.4** | **73.9** | **35.7** | **37.9** | **54.6** | **65.5**   | **67.5**  |
>
> We argue that the 5B-scale experiment should be sufficient (and matches the amount of tokens used by Fu et al.) Training with 40B tokens is incredibly expensive and a one-time effort. Here, we have a much stronger point of comparison: ProLong outperforms the Llama-3.1-8B model, which underwent long-context training by the Llama team for 800B tokens.
>
> > Misleading evaluations and final short-context task performance. Add comparisons with a llama3-8B-instruct model SFTed with UltraChat?
>
> We opted for evaluating the short-context performance before SFT because short-context tasks are heavily affected by SFT data (for example, Llama-3-8B-Instruct is better than Llama-3-8B by 4 absolute points). Our goal is to make sure that our recipe retains the base model’s performance rather than to produce a final model that competes on short-context benchmarks. Therefore, comparison is more fair and straightforward by conducting the evaluation before SFT.
>
> Regardless, we agree that it is important to report the final short-context performance of our models. We have added these results, including the ablation you suggested! Since this point was shared by other reviewers, we discuss the results in our general response above!
>
> > Longer context brings benefits due to training for 4B more tokens
>
> This must be a misunderstanding, since we very intentionally provide a comparable baseline in Table 7 (same number of tokens, learning rate, batch size, rope frequency parameter) that continues training with shorter context data for another 4B tokens, too. This is the control experiment that allows us to draw this conclusion.
>
> > Weak synthetic data generator
>
> We acknowledge this and update the paper with results with a Llama-70B generator. Our takeaways are unchanged! Please also see our general response for more details!
>
> **Questions**
>
> > Why does training on all long context data harm downstream long-context tasks?
>
> Regular pre-training datasets are incredibly diverse and known to induce a wide range of capabilities in models. Selecting only the subset of long-form documents introduces bias, as we cannot assume that this subset will have the same coverage over all types of knowledge and reasoning capabilities as regular pre-training data. Therefore, to align the distribution with the pre-training distribution and prevent catastrophic forgetting, mixing in high-quality short-context data improves performance.
>
> > Could you evaluate Prolong on RULER?
>
> Of course! We have included these results in the updated paper. Please refer to our general response.
>
> > Stage 2 batch size doubled from 4M to 8M
>
> In exploratory experiments, we observed little difference from increasing the larger batch size and our motivation for this change was to speed up training with our mini-batch reordering algorithm, which benefits from larger batch sizes, see Appendix B.3.
>
> We also doubled the batch size for the 64K-length control experiment in Table 7 (the two experiments use exactly the same hyperparameters) and still observed additional gains from 512K contexts.

---

> > ### Comment · Reviewer_kqM8 · 2024-11-22
> >
> > Thank you for your reply. My concerns are mostly addressed and I maintain my positive score.

---

### Official Review · Reviewer_ive5 · 2024-11-04

**Soundness:** 3
**Presentation:** 3
**Contribution:** 2
**Rating:** 5
**Confidence:** 5

**Summary:**

The main contributions of the paper are related to evaluations, data mix, and training recipe.

Evaluation related contribution
* Using HELMET (an existing benchmark) instead of perplexity and needle in the haystack for evaluating long context expansion.
* The authors suggest evaluating LLMs for long context after SFT instead of after pretraining.

Data mix related contributions

* Carefully designed data mixtures like high quality code and books are important.
* Additionally, the authors ablate the ratio of short context and long context data in the data mix to show that a very high fraction of long context data is detrimental to both long context and short context tasks.

Training recipe:
* Training on more tokens improves performance of long context tasks
* Increasing pretraining context length beyond target context length can also help long context tasks.
* They also show that synthetic data during SFT can be harmful to long context benchmarks

They combine these improvements to develop a new model called ProLong which achieves better results than prior models on HELMET and NoCha (another long context benchmark).

**Strengths:**

* Careful examination of individual changes (data mix, sft etc.)
* Final recipe does provide a model that outperforms some of the existing models

**Weaknesses:**

The paper contributions are somewhat limited. The main contributions are around datamixing and training recipe. The authors make some interesting claims but unfortunately don't investigate these claims further.

* The results on perplexity are a bit surprising. The authors show that increasing long context data results in perplexity improvements but long context tasks performance is lower. It would be really interesting to understand this further. Is the quality degraded due to the data mix? Or is the target eval task not correlated to the PG19 books dataset?
* The authors show that using synthetic data during SFT can lead to poor performance. It would have been helpful if the authors had verified that the generated data is correct prior to dismissing it.
* Additionally, using a large model (say Llama 70B) for synthetic data generation could have been more interesting. Perhaps, this larger model would have generated data that can be used for SFT.
* The trends between SFT evaluation and pretrained model evaluation are not clear. In Figure 2, many of the benchmarks show similar trends before and after SFT, so it's hard to conclude that it is necessary to measure performance after SFT as the authors claim.

The paper would also be stronger if the authors show that their methods generalize beyond just Llama 3 8b. Would the same set of techniques also expand context lengths effectively for other open source models?

**Questions:**

* How does this recipe scale? Would this recipe work for larger models?
* Since the model is trained for 512k context length, it would be very helpful to understand the performance of the model at 512k context length tasks. Do any of the evaluations have such long context tasks?
* The authors mention that the performance on short context tasks is important but don't report performance on these tasks for their final ProLong model. How does ProLong perform on short context tasks?

---

> ### Author Response · Authors · 2024-11-18
> **Response to Reviewer ive5**
>
> Thank you for your feedback! We are glad you recognized our careful examination of individual design decisions and highlighted our contributions across evaluation, data engineering, and the training recipe. However, there were some concerns about our claims, which we hope to address here.
>
> > The results on perplexity are surprising
>
> There have been several studies showing that perplexity often does not directly reflect model performance [1]; it is also common practice in data selection and data engineering work to trust downstream performance instead of perplexity [2][3]. Specifically, [4][5] points out that perplexity is not a good metric for long-context performance as it overlooks “key tokens” that reflect long-context abilities by averaging across all tokens. We will incorporate those additional references in the revision.
>
> [1] Liu et al., 2022. Same Pre-training Loss, Better Downstream: Implicit Bias Matters for Language Models.
>
> [2] Xie et al., 2023. DoReMi: Optimizing Data Mixtures Speeds Up Language
> Model Pretraining.
>
> [3] Li et al., 2024. DataComp-LM: In search of the next generation of
> training sets for language models
>
> [4] Fang et al., 2024. What is Wrong with Perplexity for Long-context Language Modeling?
>
> [5] Hu et al., 2024. Can Perplexity Reflect Large Language Model’s Ability in Long Text Understanding?
>
>
> > It would help if the authors had verified that the generated SFT data is correct. Additionally, using a large model (say Llama 70B) for synthetic data generation could have been more interesting.
>
> We incorporated this great suggestion for making our paper stronger. Please see our general response above! In short, using Llama 70B for synthetic data generation did not change our conclusion. We also manually inspected the generated data as we were building the synthetic data pipeline and did not spot any errors. We provide a qualitative example in the updated Table 23 and will release all our data and its generation code for transparency.
>
>
> > Trends between SFT evaluation and pretrained model evaluation are not clear
>
> We argue that in the case of long-context, the “few” tasks that show distinct trends before and after SFT are crucial for holistic evaluation and model development: (1) QA and summarization are the most prominent applications for long-context LMs, yet these models require SFT to achieve meaningful performance; (2) according to the HELMET paper, RAG—which shows distinct trends before and after SFT—best correlates with overall long-context performance. Given the negligible cost of SFT compared to long-context pre-training, performing SFT before evaluation is cost-effective and beneficial.
>
> > Results generalize beyond just Llama 3 8b?
>
> Unfortunately, we do not have the computational resources to test whether all design decisions generalize to other base models. We do not claim that our exact training configuration is optimal for all base models.
> However, our paper provides guidance on important factors to consider in simple continual long-context training---e.g., choice of evaluation, long data sources, mixing in short data, instruction tuning, tuning rope base frequency---many of which are overlooked by prior work. Finally, if we produce a long-context model with our final recipe for another base model, it would not be clear to what baselines we would compare to. For Llama-3-8b, we can compare to Llama-3.1-8b as a strong long-context baseline, which we beat in most evaluations.
>
> **Questions:**
>
> > How does this recipe scale? Would this recipe work for larger models?
>
> Full long-context training for larger models is unfortunately computationally prohibitive for us. However, the question of model scale remains intriguing, and 8B models lag behind larger models on the hardest long-context benchmarks. We hope that with hardware improvements, we can study how our recipe scales to larger models in the future.
>
> > Model performance on 512K context tasks
>
> We did not evaluate our model on 512K in the final results since (1) the maximum length that HELMET supports is 128K and (2) except MegaBeam, there is no other baseline that supports >128K context length. Regardless, we extended JsonKV, QA, and summarization in HELMET to 512K and reported the results in Table 8.
> Performance on JsonKV drops to 65% compared to 99% for 128K contexts, echoing our takeaway that one should train for longer sequences than the target length. However, the model remains useful at 512K, and the performance on QA and summarization continues increasing with longer context lengths (as shown in Table 11 and below), showing that ProLong can effectively use the extra information enabled by the longer context window.
>
> |        | 32K  | 64K  | 128K | 512K   |
> |--------|-------|-------|-------|-------|
> | QA     | 31.7  | 43.7  | 46.7  | **49.7** |
> | Summ   | 40.4  | 39.8  | 41.5  | **42.1** |
>
>
> > Final short-context task performance
>
> We have added these results, please see our discussion in the general response above!

---

> > ### Author Response · Authors · 2024-11-22
> > **Reminder for paper discussion**
> >
> > Dear reviewer,
> >
> > As we approach the end of the discussion period, we would greatly appreciate your input on the paper. We hope our responses and additional results address your concerns and welcome any further questions or suggestions.

---

> > > ### Author Response · Authors · 2024-11-27
> > > **Reminder to engage in discussion**
> > >
> > > Dear Reviewer,
> > >
> > > This is a gentle reminder regarding the paper discussion. We kindly ask if our responses and the additional results provided have addressed your concerns, and we would greatly appreciate any further questions or suggestions you may have.
> > >
> > > Thank you for your time!

---

> > > > ### Author Response · Authors · 2024-12-02
> > > > **Reminder to respond to author rebuttal**
> > > >
> > > > Dear reviewer,
> > > >
> > > > We believe that our response above addressed all your concerns and questions about our paper. As this is the last day of the discussion, we would greatly appreciate it if you could respond to our rebuttal and let us know if you have any additional question as soon as possible. Thank you!

---

> > > > > ### Comment · Reviewer_ive5 · 2024-12-02
> > > > > **Response to rebuttal**
> > > > >
> > > > > Thank you for the detailed and thoughtful response. I have updated my score based on the comments and updates to the paper.

---

### Author Response · Authors · 2024-11-18
**General Author Response**

We thank the reviewers for their careful examination of our paper. The reviewers highlighted our thorough investigation into long-context training (kqM8, DH4H), the value to the community (DH4H), the strong performance of our final model (WrRx), and the clear presentation (DH4H, WrRx). However, several of the reviewers raised similar points about additional evaluations and synthetic data generation. In response, we have revised the paper and updated the results and analysis. We believe this made the paper stronger and hope you can share our excitement!

## Additional results on RULER and InfiniteBench

We have updated the submission with final evaluations on **RULER** and **InfiniteBench** in Table 27 (also shown below). Notably, our ProLong model is **the strongest model at the 10B-scale on InfiniteBench**. On RULER, ProLong performs worse than Llama-3.1-8B but is far better than Gemini-1.5-Pro. We note that RULER and InfiniteBench have some unintuitive trends, e.g. Llama-3.1-8B also outperforms Gemini-1.5-Pro on RULER and outperforms Llama-3.1-70B on InfiniteBench. Such unintuitive trends are part of the reason why we chose HELMET. In Section 2.1, we discuss more reasons for focusing on HELMET during model development, including (1) comprehensive coverage of applications and (2) stable model-based metrics for QA and summarization tasks. More details can be found in the HELMET paper.


## Synthetic Long Instruction Data
The reviewers pointed out weaknesses in our SFT experiments, which we address in the updated paper draft. A major concern was that the model used for synthetic data generation was too weak. Therefore, we've updated the experiments in Table 24 (also shown below) with a Llama-3-70B-Instruct data generation model---as suggested by two reviewers---and confirm that our findings are unchanged. Even though using synthetic data from the 70B model is slightly better than that from the 8B model, using only short-context SFT data still performs the best. We provide a randomly-picked example of the synthetic data in Table 23 and will release the entire dataset and generation code to encourage future work. We also use a diverse set of instruction templates, which we show in Table 17.

**Why is using long synthetic SFT data not necessary?** Our finding seems to contradict previous research [1, 2]. We propose the following hypotheses at the end of Section 5: (1) previous work like [1] may have insufficient long-context training and the synthetic data acts as additional long-context training data. (2) Llama 3.1 uses SFT data that is several orders of magnitude bigger than ours and adopts multi-round SFT and preference optimization [2]---it is possible that when using an extensive short-length instruction dataset, mixing in synthetic long data avoids the model from degenerating on its long-context capabilities.

Despite our best attempts, we acknowledge “synthetic long SFT data does not help” is too strong a claim. We will revise the paper accordingly, emphasizing that “after sufficient long-context training, using only short SFT is surprisingly effective" instead.

[1] Bai et al., 2024. LongAlign: A Recipe for Long Context Alignment of Large Language Models.

[2] Llama team, 2024. The Llama 3 Herd of Models.

## Additional Short-Context Evaluations
We have added short-context performance of our final ProLong model in Table 26 (also shown below). We note that ProLong outperforms Llama-3-8B-Instruct except on MMLU and GSM8K, arguably because we do not have access to the Llama team’s extensive post-training data and its focus on math/knowledge-intensive tasks. We also incorporate reviewer kqM8’s suggestion and SFT’ed Llama-3-8B-Instruct on UltraChat for a fair comparison with ProLong. We see that ProLong achieves comparable performance with this model on short-context tasks.

---

> ### Author Response · Authors · 2024-11-18
> **Supplementary Tables**
>
> ## Additional results on RULER and InfiniteBench
> Excerpt of Table 27 from the updated submission:
>
> | Model                  | RULER | MC   | QA   | Sum  | Diag | Calc | Find  | Number | PassKey | KV    | InfiniteBench Avg.  |
> |------------------------|-------|------|------|------|------|------|-------|--------|---------|-------|-------|
> | **10B models**         |       |      |      |      |      |      |       |        |         |       |       |
> | ProLong (8B)           | 71.9  | 65.1 | 22.0 | 19.8 | 4.5  | 0.0  | 27.4  | 100.0  | 100.0   | 92.8  | **48.0**  |
> | MegaBeam-Mistral       | 78.9  | 53.7 | 18.5 | 24.8 | 12.0 | 0.0  | 24.3  | 99.7   | 100.0   | 36.4  | 41.0  |
> | Meta-Llama-3.1 (8B)    | 81.3  | 67.2 | 15.5 | 26.7 | 23.0 | 0.0  | 33.1  | 99.5   | 100.0   | 55.0  | 46.7  |
> | **Larger models**      |       |      |      |      |      |      |       |        |         |       |       |
> | Meta-Llama-3.1 (70B)   | 75.8  | 75.5 | 23.3 | 31.3 | 18.0 | 0.0  | 43.1  | 99.7   | 100.0   | 2.6   | 43.7  |
> | Gemini-1.5-Pro         | 65.3  | 77.5 | 27.7 | 29.0 | 97.5 | 0.0  | 58.0  | 100.0  | 100.0   | 70.4  | 62.2  |
>
> _Evaluation on RULER and InfiniteBench at a context length of 128K_
>
> ## Synthetic Long Instruction Data
> Table 24 from the updated submission:
>
> | % Synthetic Data   | JsonKV*        | RAG   | Re-rank | ICL   | QA*   | Summ.* | Avg.  |
> |--------------------|----------------|-------|---------|-------|-------|--------|-------|
> | **0%**               | 65.7          | 58.1  | 38.5    | 80.3  | 49.7  | 42.1   | **55.7** |
> | 1% (from 8B)       | 61.5          | 57.0  | 38.3    | 80.8  | 45.3  | 41.5   | 54.1  |
> | 1% (from 70B)      | 64.7          | 57.3  | 37.4    | 78.4  | 47.0  | 40.8   | 54.2  |
> | 3% (from 8B)       | 62.0          | 56.4  | 37.9    | 80.6  | 44.8  | 39.5   | 53.5  |
> | 3% (from 70B)      | 65.7          | 57.4  | 38.0    | 80.1  | 48.7  | 42.5   | 55.4  |
> | 10% (from 8B)      | 70.3          | 55.5  | 36.1    | 80.6  | 41.7  | 39.4   | 53.9  |
> | 10% (from 70B)     | 66.3          | 57.0  | 33.4    | 81.2  | 45.3  | 38.4   | 53.6  |
> | 50% (from 8B)      | 45.8          | 48.8  | 18.8    | 70.5  | 42.3  | 33.3   | 43.3  |
> | 50% (from 70B)     | 55.8          | 53.9  | 23.5    | 74.1  | 50.7  | 39.9   | 49.7  |
> _Evaluation on HELMET at 32K and 64K length except tasks with *, which are evaluated at 512K for stress testing._
>
> ## Additional Short-Context Evaluations
> Table 26 from the updated submission:
>
> | Model                                | HellaSwag | MMLU  | ARC-c | WinoGrande | GSM8K | Avg.  |
> |--------------------------------------|-----------|-------|-------|------------|-------|-------|
> | Llama-3-8B + Fu et al. (2024) | 82.5      | 63.9  | 63.6  | 75.1       | 42.2  | 65.4  |
> | Llama-3-8B                           | 82.1      | 66.5  | 59.4  | 77.1       | 44.7  | 66.0  |
> | Llama-3-8B-Instruct + UltraChat      | 82.1      | 65.1  | 64.3  | 75.5       | 60.7  | 69.5  |
> | Llama-3-8B-Instruct                  | 78.5      | 67.0  | 60.8  | 74.2       | 68.5  | 69.8  |
> | *ProLong*                            | 82.8      | 64.6  | 64.7  | 76.2       | 58.9  | 69.4  |

---

### Meta-Review · Area_Chair_881K · 2024-12-21

**Metareview:**

The paper presents ProLong-8B, a language model fine-tuned to handle extended context lengths up to 512K tokens. Key findings include the effectiveness of combining code repositories and books with high-quality short data, the benefits of training with sequence lengths exceeding evaluation lengths, and the strong performance achieved using only short instruction datasets. However, there are major concerns regarding the experiments and their results i.e. how these results can be interpreted as valuable insights for the community, the intuitions behind some observations are not clearly justified, and comparison with relevant baselines. I encourage the authors to address these issues, focusing on how their results can be translated as more generalisable insights for the community.

**Additional Comments On Reviewer Discussion:**

n/a

---

### Decision · Program_Chairs · 2025-01-22

Reject